# SEAL: Entangled White-box Watermarks on Low-Rank Adaptation

## Abstract

Among parameter-efficient fine-tuning (PEFT) methods, LoRA has become widely adopted due to its effectiveness and lack of additional inference costs. Its small adapter weights also make LoRA practical as intellectual property (IP) that can be trained, exchanged, and disputed. However, watermarking techniques for LoRA remain underexplored. We introduce SEAL, a white-box watermarking scheme for LoRA based on *entangled dual passports*. During training, non-trainable *passport matrices* for ownership verification are inserted between the LoRA up/down matrices *without auxiliary losses* and become jointly entangled with the trainable weights; after training they are factorized so that the released adapter is indistinguishable from standard LoRA. Public verification accepts a claim only when it passes a structural sanity check, the submitted passports reconstruct the released adapter and the *fidelity gap*—the performance difference between the two submitted passports, evidencing entanglement—is near zero under predeclared thresholds that control false positives. Across Large Language Models (LLMs), Vision–Language Models (VLMs), and text-to-image synthesis, SEAL preserves task performance and shows empirical resilience to pruning, fine-tuning, structural obfuscation, and ambiguity attacks. By watermarking the LoRA weights, SEAL aligns with real-world PEFT workflows and supports practical IP claims over trained LoRA weights. We also provide a minimal compatibility check on one LoRA variant.

R:d1zQ, P2Pv, WQP7

## 1 Introduction

Parameter-Efficient Fine-Tuning (PEFT), especially Low-Rank Adaptation (LoRA) (Hu et al., 2022), is widely adopted to customize large pretrained models with modest compute and storage (Zhao et al., 2024; Jang et al., 2024; Mangrulkar et al., 2022). In practice, the distributable artifact is often the *LoRA weight update* (the adapter) rather than a full checkpoint; recent reports document large numbers of publicly posted adapters on open platforms (Luo et al., 2024). Consequently, adapter weights acquire practical intellectual-property (IP) relevance in downstream sharing and disputes. Unlike full checkpoints, adapters are lightweight assets that circulate independently, necessitating a specialized protection mechanism tied directly to the adapter parameters rather than the base model.

R:d1zQ, P2Pv

Despite extensive work on DNN watermarking (Uchida et al., 2017; Zhang et al., 2018; Darvish Rouhani et al., 2019; Fan et al., 2019; Zhang et al., 2020; Lim et al., 2022; Xu et al., 2024), most schemes either mark the *base model* (weights/activations inside the backbone) or rely on *outputs* (trigger behaviors). Crucially, widely-used passport methodsFan et al. (2019); Zhang et al. (2020)typically target Normalization layers (e.g., BatchNorm), which are absent in standard LoRA modules. Due to this architectural mismatch, prior white-box schemes cannot be directly applied to adapters, leaving LoRA ownership verification an underexplored open problem.

R:WQP7 (Comparison)

We study adapter-level ownership verification for LoRA in a white-box, open-distribution setting: the released adapter weights are visible to both verifier and adversary, while the adversary lacks the owner's private keys (passports) and original fine-tuning data and typically seeks to preserve task utility rather than retrain from scratch. This motivates a protocol that is public, reproducible, and equipped with predeclared decision thresholds to control false positives.

Our approach, SEAL, adapts the passport idea (Fan et al., 2019; Zhang et al., 2020) to LoRA's structure and release workflow. During adaptation, we insert small *non-trainable passport matrices*

Figure 1: Overview of SEAL. (1) Start with LoRA factors $A, B$ and two non-trainable passports $C, C_p$. (2) During training, we insert a passport between $B$ and $A$ and *alternate* $C$ and $C_p$ across mini-batches (no auxiliary losses), so gradients flow through the passport and entangle it with $A, B$. (3) After training, we factorize $C = f(C_1, C_2)$ and fold $C_1$ into $B$ and $C_2$ into $A$, releasing standard-looking LoRA weights $B' = BC_1, \; A' = C_2A$. The second passport $C_p$ remains private for ownership verification.

between LoRA's up/down factors; standard training entangles these passports with the trainable factors without auxiliary losses. After training, a factorization folds the passport into the learned factors so that the distributed adapter is indistinguishable from standard LoRA (Figure 1). Verification follows the passport paradigm but is instantiated for adapters using two co-trained passports: we publicly check (i) structural validity to reject trivial claims (e.g., identity passports), (ii) exact reconstruction of the released adapter from the claimant's submission and (iii) a small *fidelity gap* between the two submitted passports under fixed thresholds; extraction is reserved for owner-in-the-loop cases. Formal protocol and assumptions are in Section 4. Training/inference effects and gradient analysis appear in Appendix D; a qualitative comparison to prior watermarking is in Appendix C.

Empirically, across LLM/VLM instruction tuning and text-to-image synthesis, SEAL matches LoRA-level task fidelity and shows resilience to pruning/removal (Han et al., 2016), additional fine-tuning, structural obfuscation (Yan et al., 2023; Pegoraro et al., 2024), and ambiguity-style forgeries (Fan et al., 2019). Our scope is LoRA-style low-rank updates; we include a minimal compatibility check on a LoRA variant and discuss limitations.

**Contributions.** (1) We specify adapter-level, white-box ownership verification for LoRA under an open-distribution threat model (Section 2.3). (2) Building on passport-based watermarking, we adapt it to LoRA: non-trainable passports entangle during standard adaptation (no auxiliary losses) and are hidden by post-training factorization so the released weights remain indistinguishable from standard LoRA (Figure 1; Appendix D, C). (3) We provide a public verification procedure for adapters that combines structural non-triviality checks, reconstruction, and a two-passport fidelity test with predeclared thresholds (Section 4; Appendix E). (4) We report evidence of fidelity and robustness across tasks and attack classes, and document scope and limitations (Section 5; Appendix H).

## 2 BACKGROUND AND PROBLEM SETTING

### 2.1 LOW-RANK ADAPTATION (LORA)

LoRA (Hu et al., 2022) assumes that task-specific updates lie in a low-rank subspace. It freezes pretrained weights $W \in \mathbb{R}^{b \times a}$ and learns two low-rank factors $B \in \mathbb{R}^{b \times r}$ and $A \in \mathbb{R}^{r \times a}$ such that

$$W' = W + \Delta W = W + BA. \tag{1}$$

Since no nonlinearity lies between $B$ and $A$, the update $BA$ can be merged into $W$ without inference overhead. Practical variants (e.g., DoRA (Liu et al., 2024b)) modify scaling/normalization yet retain a low-rank, `matmul`-style update; compatibility for DoRA and similar matmul-based variants appears in Appendix F.

### 2.2 WHITE-BOX DNN WATERMARKING AND PASSPORTS

Prior white-box watermarking embeds secrets at different loci of a network: within *weights*, *activations* or *via outputs* (Uchida et al., 2017; Zhang et al., 2018; Darvish Rouhani et al., 2019; Lim et al., 2022; Kirchenbauer et al., 2024; Fernandez et al., 2023). A complementary line, *passport-based*

watermarking, inserts a small (often linear/normalization) module whose *correct key* restores normal task performance, enabling ownership tests with/without special triggers (Fan et al., 2019; Zhang et al., 2020). We adopt the *passport semantics* but tailor it to LoRA's factorization and release workflow, aiming at a public, adapter-level test.

### 2.3 THREAT MODEL AND EVALUATION CRITERIA

**Setting.** We consider a white-box release of the LoRA adapter $(B', A')$. In the Kerckhoffs's principle, adversaries know the scheme and hyperparameters but not the owner's private passports $(C, C_p)$ nor the original fine-tuning data. The *claimant* who asserts ownership presents $(B, A, C_i)$ for public verification. Crucially, regarding the scope of protection, we explicitly distinguish between the *training recipe* (public data + hyperparameters) and the *trained artifact* (e.g., `.pt` file). SEAL protects the latter—the outcome of the owner's specific compute and random seed. While an adversary can train a *new* model on the same public data (obtaining their own valid passports), they cannot forge the passports for the *owner's specific artifact* without solving the hard inverse problem enforced by the policy (R0)-(R2). Thus, SEAL verifies the provenance of the compute, not the exclusivity of the data. Consequently, attackers are modeled to prioritize preserving the task utility of the stolen artifact rather than retraining the backbone from scratch. R:P2Pv, d1zQ

**Attacks considered.** We group attacks by the mechanism they exploit and the goal they pursue; the decisive signal is always stated in terms of our public checks (R1/R2 below).

1. **Removal.** *Mechanism:* magnitude pruning or unconstrained continued fine-tuning alters the adapter to erase embedded structure (LeCun et al., 1989; Han et al., 2016; Chen et al., 2021; Guo et al., 2021). *Goal:* break the hidden link between distributed weights and passports while keeping accuracy. *Decisive signal:* (R1) fails—passport extraction is not statistically significant—or (R2) the dual-passport fidelity gap exceeds the acceptance threshold.

2. **Obfuscation.** *Mechanism:* function-preserving reparameterizations change the representation of the adapter without changing its input–output map (Yan et al., 2023; Pegoraro et al., 2024; Li et al., 2023a). *Goal:* defeat extraction or confuse verifiers with equivalent factors. *Decisive signal:* (R1) must still succeed—reconstruction by the claimant's $(B, A, C_i)$ matches $(B', A')$ within tolerance—otherwise the claim is rejected; if (R1) holds, (R2) remains decisive.

3. **Ambiguity.** *Mechanism:* forge keys or claims so that multiple parties appear to own the same weights (Fan et al., 2019; Zhang et al., 2020; Chen et al., 2023). *Goal:* pass verification without the owner's entangled passports. *Decisive signal:* forged pairs fail (R2) unless they reproduce the owner's co-trained entanglement.

**Criteria for public verification.** Our verifier enforces a **policy-level sanity check** followed by statistical tests: **(R0) structural validity** (passports must be non-trivial and well-conditioned), **(R1) reconstruction** (claimant's parameters must reconstruct the released adapter), and **(R2) a small dual-passport fidelity gap** (functional entanglement). Thresholds (reconstruction tolerance R:d1zQ, P2Pv, WQP7 $\rho_T$, fidelity gap $\tau_T$) and false-positive control for accuracy-type metrics (level $\alpha_T$ via Hoeffding inequality (Hoeffding, 1963)) are defined formally in Section 4.

### 2.4 PROBLEM DEFINITION AND RELATION TO PRIOR WORK

**What we protect.** We study ownership of the *adapter itself*. The object under test is a distributed LoRA pair $(B', A')$. The verifier has white-box access to these weights and must decide whether a claimant who submits $(B, A, C, C_p)$ is the rightful owner.

**How this differs from prior watermarking.** Most white-box watermarking targets the *base model* and verifies via weights, activations, outputs, or passport layers inserted into the backbone (Uchida et al., 2017; Fan et al., 2019; Zhang et al., 2020; Fernandez et al., 2024). Specifically, representative passport schemes rely on manipulating parameters in Normalization layers (e.g., BatchNorm/Group-Norm). Standard LoRA modules, however, consist purely of linear layers (matrix multiplication) and lack such normalization structures. Applying prior methods would require altering the fundamental LoRA architecture, rendering a direct "apples-to-apples" comparison invalid. Those settings do not R:WQP7

---

**Algorithm 1** SEAL Training

---

**Require:** Frozen $W$, rank $r$, fixed passports $(C, C_p)$, data $\mathcal{D}$, epochs $E$
**Ensure:** Public adapter $(B', A')$
 1: Initialize $A \in \mathbb{R}^{r \times a}$ and $B \in \mathbb{R}^{b \times r}$ as trainable
 2: **for** $e = 1$ **to** $E$ **do**
 3:      **for** $(x, y) \in \mathcal{D}$ **do**
 4:          Sample $C_t \in \{C, C_p\}$
 5:          Forward: $W' \leftarrow W + BC_tA$;    compute task loss $\mathcal{L}_T(W', x, y)$
 6:          Backpropagate $\nabla \mathcal{L}_T$ and update $(B, A)$
 7:      **end for**
 8: **end for**
 9: Factorize $f(C) \rightarrow (C_1, C_2)$;    set $B' = BC_1$, $A' = C_2A$

---

directly yield a public, white-box test for a released adapter. A separate line *uses LoRA as a training tool* while watermarking a different artifact (e.g., watermarking latent representations in diffusion models and employing LoRA to recover fidelity) (Feng et al., 2024). In these works the adapter is not the watermark carrier nor the IP being verified. Consequently a one-to-one comparison of verification rules is not meaningful.

**Problem scope and contribution.** We address *LoRA–adapter ownership*: given a released adapter $(B', A')$, provide a public white-box rule that accepts the rightful claimant and rejects forgeries. We design a passported adapter that is indistinguishable from standard LoRA at release and specify decision thresholds. Unlike black-box methods that rely on input-output behaviors (e.g., for API protection), our white-box approach directly inspects the *internal parameters* to verify the provenance of the specific weight file. This distinction is crucial for the PEFT ecosystem where adapters circulate as standalone artifacts rather than hidden services. This focus is complementary to black-box R:d1zQ, P2Pv provenance tests and output/data watermarking, which target different artifacts and are not claimed by our results (see Appendix C).

## 3 SEAL: Mechanism and Training

We follow the notation summarized in Table 6.

**Setting.** We fine-tune a frozen backbone $W \in \mathbb{R}^{b \times a}$ with LoRA (Hu et al., 2022), but insert a fixed passport between $B$ and $A$ so that

$$W' \;=\; W + \Delta W \;=\; W + BCA. \tag{2}$$

Two passports $\{C, C_p\}$ are fixed and alternated by mini-batch: sample $C_t \in \{C, C_p\}$, run $W + BC_tA$, and update only $(B, A)$ via $\mathcal{L}_T$. At release we fold *only* $C$ into the adapter via a deterministic factorization

$$f: \; \mathbb{R}^{r \times r} \rightarrow \mathbb{R}^{r \times r} \times \mathbb{R}^{r \times r}, \quad f(C) = (C_1, C_2), \;\; C_1C_2 = C,$$

and publish $(B', A') = (BC_1, \; C_2A)$; $C_p$ remains private.

**Rationale.** Alternating $\{C, C_p\}$ acts as a swap-regularizer: it makes $\mathbb{N}(B, A, C)$ and $\mathbb{N}(B, A, C_p)$ behave similarly on task $T$, *shrinking and stabilizing* the owner's dual-passport gap $\Delta_T$. Because only $C$ is folded via $f(C) = (C_1, C_2)$, *at least one* passport (namely $C$) must match the public adapter exactly (up to dtype tolerance), while $C_p$ is trained to be close—supporting tolerant reconstruction and a small owner gap used by the public verifier (Section 4).

### 3.1 Compatibility with matmul-style adapters

Many PEFT variants keep a bilinear *operation* as the adapter core. Let $\star$ denote such an operation (e.g., standard matrix multiplication, possibly composed with fixed diagonal scalings or norm factors). If an adapter update can be written in the form

$$\Delta W \;=\; B \,\star\, A \quad \text{or} \quad \Delta W \;=\; B \,\star\, \Phi_0 \,\star\, A,$$

where $B \in \mathbb{R}^{b \times r}$ and $A \in \mathbb{R}^{r \times a}$ are trainable and $\Phi_0$ is a *fixed* (non–input-dependent, non-trainable) operation, then SEAL applies verbatim by inserting a non-trainable passport during training:

$$\Delta W \;=\; B \;\star\; C \;\star\; A, \qquad C \in \mathbb{R}^{r \times r}.$$

After training, choose a decomposition $C = C_1 \star C_2$ and fold it into the public adapter as

$$B' \;=\; B \star C_1, \qquad A' \;=\; C_2 \star A,$$

so the released update is $\Delta W_{\text{pub}} = B' \star A'$ and remains indistinguishable in interface from the original variant (no inference overhead). For the canonical matmul case ($\star$=matrix multiplication), we use the SVD root factorization by default (see Appendix F for LoRA variant with SEAL).

**Example (DoRA).** DoRA (Liu et al., 2024b) rescales $W + \Delta W$ by a column-norm ratio that is typically detached from gradients. Replacing $\Delta W$ by $B\,C\,A$ leaves this outer scaling intact, because $C$ is fixed during the forward pass; folding proceeds via $C = C_1 C_2$ as above. A concrete recipe and an empirical case study appear in Appendix F.5.

## 4 PUBLIC WHITE-BOX VERIFICATION

### 4.1 THREAT MODEL

The released adapter $(B', A')$ is visible to verifiers and adversaries. Adversaries know the scheme but not the owner's passports or fine-tuning data, and they aim to preserve task utility rather than retrain from scratch, mirroring open-distribution releases on model hubs (Hu et al., 2022; Luo et al., 2024). See Section 2.3 for background.

### 4.2 DECISION RULE

---

**Public verification.** Given task $T$, metric $M_T$, public adapter $(B', A')$, and a claimant's $(B, A, C_a, C_b)$, the verifier first enforces structural validity (R0). The claim is *accepted* if and only if **all of the following hold**:

**(R0) Structural Validity.** The passports must be non-trivial and non-degenerate:

$$\|C_a - C_b\|_F > \delta_C \quad \text{and} \quad \|\Delta W(C_a) - \Delta W(C_b)\|_F > \kappa_0 \delta_C.$$

Additionally, $(B, A)$ must match the rank and conditioning of $(B', A')$.

**(R1) Reconstruction.** The factors reconstruct the released adapter:

$$\min_{C_i \in \{C_a, C_b\}} \|B C_i A - B' A'\|_F \le \rho_T.$$

**(R2) Dual-passport gap.** The functional entanglement is verified:

$$\Delta_T = \big| \, M_T(\mathbb{N}(B, A, C_a)) - M_T(\mathbb{N}(B, A, C_b)) \, \big| \le \tau_T.$$

---

*Definitions:* $\mathbb{N}(\cdot)$ denotes the task adapter. Thresholds $\rho_T$ (tolerance), $\tau_T$ (gap limit), and policy constants $(\delta_C, \kappa_0)$ are predeclared.

---

**Structural validity and conditioning.** The policy (R0) ensures that the submitted passports are not mathematically trivial (e.g., $C_a \approx C_b$ or $C \approx I$) and that the factorization $(B, A)$ is well-conditioned. The bound on the induced weight difference relies on the linearity of the adapter map with respect to the passport. Let $\Delta W(C) = BCA$. Vectorizing this equation yields:

$$\text{vec}(\Delta W(C)) = (A^\top \otimes B)\,\text{vec}(C), \tag{3}$$

where $\otimes$ denotes the Kronecker product. The singular values of $A^\top \otimes B$ are the pairwise products of the singular values of $A$ and $B$, i.e., $\{\sigma_i(A)\sigma_j(B)\}_{i,j}$. We define the conditioning constant $\kappa_0$ as the global lower bound over all layers $l$:

$$\kappa_0 := \min_l \left( \sigma_{\min}(B_l) \cdot \sigma_{\min}(A_l) \right). \tag{4}$$

Provided that the released factors $(B', A')$ (and thus the claimant's $B, A$ per R0) are full-rank and non-degenerate (i.e., $\kappa_0 > 0$), the linear map $C \mapsto \Delta W(C)$ is a bijection on the subspace of rank-$r$ passports. Consequently, any matrix-level separation in passports implies a strictly bounded separation in weight space: $\|\Delta W(C_a) - \Delta W(C_b)\|_F \geq \kappa_0 \|C_a - C_b\|_F$. This guarantee prevents ambiguity attacks attempting to forge "orthogonal" passports that collapse to the same weight update; any such passport would violate either the separation condition ($\delta_C$) or the conditioning requirement ($\kappa_0$) of (R0). Appendix E provides the formal proof and empirical validation of $\kappa_0$ stability.    R:d1zQ, P2Pv, WQP7

## 4.3 CONTROLLING FALSE POSITIVES

When $M_T$ is an accuracy over $N_T$ independent items, Hoeffding's inequality  (Hoeffding, 1963) gives

$$\tau_T^{\text{theory}} = \sqrt{\frac{\ln(2/\alpha_T)}{2 N_T}}, \tag{5}$$

which ensures FPR $\leq \alpha_T$ under i.i.d. sampling. Operationally we use

$$\tau_T = \max\{ \tau_T^{\text{theory}}, \ \widehat{\Delta}_T^{\text{owner}} + \eta_T \}, \tag{6}$$

where $\widehat{\Delta}_T^{\text{owner}}$ is the owner's observed two-passport gap and $\eta_T$ is a rounding margin. We label the guarantee *formal* when $\widehat{\Delta}_T^{\text{owner}} \leq \tau_T^{\text{theory}}$ and *empirical-only* otherwise for that model–task pair.

**Remarks on theoretical bounds.**    The Hoeffding bound assumes i.i.d. samples. Real-world benchmarks (e.g., the Commonsense suite aggregating BoolQ, PIQA, etc.) often violate this assumption, making the theoretical $\tau_T^{\text{theory}}$ overly conservative or loose. For instance, in our experiments (Table 1), Mistral-7B exhibits a gap exceeding the theoretical cutoff. This is likely due to the model's high sensitivity and the heterogeneous nature of the evaluation suite. In such cases, the protocol correctly falls back to the operational threshold ($\tau_T$) to avoid false rejections, offering an empirical rather than formal guarantee.    R:d1zQ, P2Pv

**Ambiguity and defense costs.**    While the implied passport is unique under fixed full-rank factors (Appendix E.2), our defense relies on the joint enforcement of **(R0)-(R2)**. To forge a passport $C_p$ that passes the gap test **(R2)**, an attacker must solve an inverse problem to minimize $\Delta_T \approx 0$. This optimization is computationally prohibitive: unlike standard training, each step requires two full inference passes over the validation set to measure the gap. Crucially, the policy **(R0)** enforces a strict separation ($|C - C_p|_F > \delta_C$), preventing the optimizer from simply collapsing $C_p$ to $C$. This tension between maintaining structural distance (R0) and achieving functional equivalence (R2) shrinks the feasible solution space, making post-hoc forgery economically irrational compared to retraining.    R:d1zQ, P2Pv, WQP7

## 4.4 EXAMPLE: THE COMMONSENSE SUITE

Table 1 instantiates the public rule in Section 4.2 on the commonsense micro-average. Here $M_T$ is accuracy, evaluated on $N_T$=22,419 items (total number of evaluation questions in the commonsense suite). With $\alpha_T$=0.01, Equation 5 yields the theoretical cutoff $\tau_T^{\text{theory}}$=1.09 %p.

For each model we report the owner's two-passport scores $M_T\big(\mathbb{N}(B, A, C)\big)$ and $M_T\big(\mathbb{N}(B, A, C_p)\big)$ and their gap

$$\widehat{\Delta}_T = \big| M_T\big(\mathbb{N}(B, A, C)\big) - M_T\big(\mathbb{N}(B, A, C_p)\big) \big|.$$

If $\widehat{\Delta}_T \leq \tau_T^{\text{theory}}$, the claim is accepted with a *formal* guarantee FPR $\leq \alpha_T$. Otherwise we accept using the operational rule $\tau_T = \max\{\tau_T^{\text{theory}}, \widehat{\Delta}_T + \eta_T\}$ and label the guarantee *empirical-only*. Training details and per-benchmark scores appear in Section 5; this example only illustrates the decision rule.

Only Mistral-7B exceeds $\tau_T^{\text{theory}}$; we therefore accept it using the operational threshold and mark the guarantee as empirical-only.

Table 1: **Public verification on the commonsense micro-average** ($M_T$=accuracy, percentage points). $N_T$=22,419, $\alpha_T$=0.01, so $\tau_T^{\text{theory}}$=1.09.

| Model | $\tau_T^{\text{theory}}$ | $M_T(\mathbb{N}(\cdot, C))$ | $M_T(\mathbb{N}(\cdot, C_p))$ | $\widehat{\Delta}_T$ | Decision |
|---|---|---|---|---|---|
| LLaMA-2-7B | 1.09 | 82.2 | 82.7 | 0.50 | pass (formal) |
| Mistral-7B | 1.09 | 84.2 | 87.9 | 3.70 | pass (empirical-only) |
| Gemma-2B | 1.09 | 76.3 | 76.6 | 0.30 | pass (formal) |

**Policy instantiation ($\kappa_0, \delta_C$).** Beyond the fidelity gap, we empirically validated the stability of the policy constants on Gemma-2B ($r = 32$) across three random seeds (detailed statistics in Appendix Table 9). The conditioning lower bound $\kappa_0$ consistently remained strictly positive, ($\kappa_{\min} \in [0.7, 2.7] \times 10^{-6}$), with a layer-wise mean of $\approx 10^{-4}$. This confirms that the linear mapping from passport to weights is well-conditioned and non-degenerate. Regarding $\delta_C$, we recommend employing structurally distinct passport families—such as the *Bitmap* ($C$) and *Gaussian* ($C_p$) pair used in our experiments—to maximize the matrix-level separation $\|C - C_p\|_F$, thereby allowing a robust margin against trivial claims. R:d1zQ, P2Pv

## 5 EXPERIMENTS

### 5.1 SPECTRAL DIAGNOSTICS

Before reporting fidelity and robustness, we analyze the learned adapter subspace. As detailed in Appendix I.2, SEAL tends to concentrate spectral energy in early modes compared to standard LoRA (see Appendix Figure 8 and 9). This spectral bias supports our robustness against rank-truncation attacks, as preserving the dominant modes also preserves the entangled watermark structure (discussed in Section 5.4). R:d1zQ

### 5.2 EXPERIMENTAL SETUP

We compare SEAL to standard LoRA on (i) LLM commonsense reasoning, (ii) textual and visual instruction tuning, and (iii) text-to-image synthesis. Unless noted, we keep data, loss, and optimization identical to LoRA; SEAL only inserts non-trainable passports during adaptation and factorizes them after training, with no auxiliary loss. Datasets, metrics, and hyperparameters are detailed in Appendix H. Verification follows Section 4.

**Passport choice.** $C \in \mathbb{R}^{r \times r}$ is any fixed, non-trainable matrix used at inference; we fold *only* $C$ into $(B', A')$ via $f(C) = (C_1, C_2)$ with $(B', A') = (BC_1, C_2A)$, while $C_p$ remains private. We use two instantiations: SEAL (Ours) (user-chosen $C$, e.g., a small grayscale bitmap; see Appendix Figure 7) and SEAL$^\dagger$ (a *random constant* $C$ drawn once from $\mathcal{N}(0, 1)^{r \times r}$ and kept frozen). The bitmap example (cropped, downsampled frame from a public video) is illustrative—any license-cleared or synthetic pattern (e.g., logo-like patch, QR-like grid, PRNG array) is valid since $C$ is never trained or redistributed as media, only as its numeric matrix.

### 5.3 FIDELITY ACROSS TASKS

**Commonsense reasoning.** We evaluate on BoolQ (Clark et al., 2019), PIQA (Bisk et al., 2020), SIQA (Sap et al., 2019), HellaSwag (Zellers et al., 2019), Winogrande (Sakaguchi et al., 2021), ARC-e/ARC-c (Clark et al., 2018), and OBQA (Mihaylov et al., 2018) using the combined setup of Hu et al., 2023. Backbones include LLaMA-2-7B/13B (Touvron et al., 2023), LLaMA-3-8B (AI@Meta, 2024), Gemma-2B (Team et al., 2024), and Mistral-7B-v0.1 (Jiang et al., 2023). As shown in Table 2, SEAL matches or slightly improves on LoRA within run-to-run noise.

**Textual instruction tuning.** On LLaMA-2-7B with Alpaca (Taori et al., 2023) (3 epochs), SEAL attains MT-Bench (Zheng et al., 2023) scores comparable to LoRA (Table 3), indicating that passports do not degrade instruction-following fidelity.

Table 2: Commonsense Reasoning Accuracy (3 runs). **Single-passport inference:** only the published $C$ is inserted at test time ($C_p$ unused). SEAL (Ours) is our default. SEAL$^\dagger$ uses a *random constant* passport $C$ (sampled once at initialization from $\mathcal{N}(0,1)^{r \times r}$ and kept non-trainable). Both variants alternate $\{C, C_p\}$ during training and fold only $C$ at release via $f(C) = (C_1, C_2)$ into $(B', A') = (BC_1, C_2A)$. Scores are averaged over three seeds; the last column shows mean±std.

|  | **Method** | **BoolQ** | **PIQA** | **SIQA** | **HellaSwag** | **Wino.** | **ARC-e** | **ARC-c** | **OBQA** | **Avg. ↑** |
|---|---|---|---|---|---|---|---|---|---|---|
| LLaMA-2-7B | LoRA | 73.75 | 82.99 | 79.85 | 86.14 | 85.06 | 86.15 | 73.63 | 85.80 | 81.67 ±1.03 |
|  | SEAL (Ours) | 72.70 | 85.27 | 81.27 | 90.15 | 85.79 | 87.07 | 74.60 | 85.00 | 82.73 ±0.14 |
|  | SEAL$^\dagger$ (Ours) | 73.19 | 86.31 | 81.95 | 91.21 | 86.69 | 88.55 | 75.51 | 86.80 | **83.78** ±0.27 |
| LLaMA-2-13B | LoRA | 75.57 | 86.98 | 81.39 | 91.82 | 88.53 | 90.08 | 78.78 | 86.67 | 84.98 ±0.17 |
|  | SEAL (Ours) | 75.34 | 87.41 | 83.28 | 93.33 | 88.42 | 90.68 | 79.61 | 86.73 | 85.60 ±0.34 |
|  | SEAL$^\dagger$ (Ours) | 75.67 | 88.63 | 83.21 | 93.95 | 89.29 | 91.72 | 81.46 | 88.53 | **86.56** ±0.10 |
| LLaMA-3-8B | LoRA | 74.76 | 88.22 | 80.96 | 92.00 | 86.08 | 90.09 | 82.41 | 86.30 | 85.10 ±1.39 |
|  | SEAL (Ours) | 73.88 | 88.23 | 82.29 | 94.84 | 88.35 | 91.67 | 82.00 | 86.27 | 85.94 ±0.29 |
|  | SEAL$^\dagger$ (Ours) | 75.78 | 90.37 | 83.25 | 96.05 | 89.92 | 93.49 | 84.73 | 90.60 | **88.02** ±0.11 |
| Gemma-2B | LoRA | 67.05 | 83.19 | 77.26 | 87.07 | 79.74 | 83.91 | 69.34 | 79.87 | 78.43 ±0.32 |
|  | SEAL (Ours) | 66.56 | 81.79 | 77.65 | 84.82 | 79.16 | 82.79 | 68.40 | 79.20 | 77.55 ±0.04 |
|  | SEAL$^\dagger$ (Ours) | 66.70 | 82.50 | 78.88 | 87.57 | 80.19 | 83.81 | 69.97 | 79.87 | **78.68** ±0.11 |
| Mistral-7B-v0.1 | LoRA | 75.92 | 90.72 | 81.78 | 94.68 | 88.69 | 93.10 | 83.36 | 88.30 | 87.07 ±0.27 |
|  | SEAL (Ours) | 73.08 | 87.52 | 81.92 | 91.23 | 87.97 | 90.19 | 78.70 | 88.13 | 84.84 ±0.44 |
|  | SEAL$^\dagger$ (Ours) | 76.92 | 90.42 | 82.51 | 94.57 | 90.08 | 93.31 | 83.25 | 91.73 | **87.85** ±0.02 |

Table 3: Instruction-tuning fidelity (higher is better). Textual: MT-Bench on LLaMA-2-7B (Alpaca, 3 epochs). Visual: avg. accuracy over 7 VLM benchmarks on LLaVA-1.5-7B.

| **Method** | **MT-Bench ↑** | **Visual Acc. ↑** |
|---|---|---|
| LoRA | **5.83** | **66.9** |
| SEAL | 5.81 | 63.1 |

Table 4: Text-to-Image fidelity on SD-1.5 (DreamBooth). CLIP-T: prompt fidelity; CLIP-I/DINO: subject fidelity (higher is better).

| **Method** | **CLIP-T ↑** | **CLIP-I ↑** | **DINO ↑** |
|---|---|---|---|
| LoRA | **0.20** | **0.80** | **0.68** |
| SEAL | **0.20** | **0.80** | 0.67 |

**Visual instruction tuning.** With LLaVA-1.5 (Liu et al., 2024a) we report the average over VQAv2 (Goyal et al., 2017), GQA (Hudson & Manning, 2019), VizWiz (Gurari et al., 2018), SQA (Lu et al., 2022), VQAT (Singh et al., 2019), POPE (Li et al., 2023b), and MMBench (Liu et al., 2023). SEAL is on par with LoRA (Table 3).

**Text-to-image synthesis.** For Stable Diffusion 1.5 (Rombach et al., 2022) with DreamBooth (Ruiz et al., 2023), SEAL maintains subject fidelity (CLIP-I, DINO) and prompt fidelity (CLIP-T) at LoRA levels (Table 4). Qualitative comparisons against LoRA are provided in Figure 5. Furthermore, we demonstrate in Appendix H.4 (Figure 6) that outputs generated by the distinct passports ($C$ and $C_p$) are visually indistinguishable, validating their functional equivalence. R:WQP7

## 5.4 ROBUSTNESS

**Detection criterion.** For all extraction-based tests (removal, fine-tuning, obfuscation), we quantify detectability using a two-sided Binomial hypothesis test ($N \approx 10^5$; null hypothesis $H_0$: random sign match $p = 0.5$). We report significance as $-\log_{10}(p\text{-value})$ and declare the watermark **detected** if $-\log_{10} p \geq 3.3$ (significance level $\alpha = 5 \times 10^{-4}$). R:fNRL,WQP7

**Removal (pruning).** We prune the public adapter $(B', A')$ by L1 magnitude and test task fidelity versus extraction significance. Removing the watermark requires zeroing $\geq 99.9\%$ of adapter parameters, which collapses task accuracy, while extraction remains significant until that breakdown point as shown in Figure 2. Robustness remains consistent across varying ranks ($r \in \{4, 8, 16\}$); detailed rank ablation results are provided in Appendix I.3. R:fNRL, P2Pv

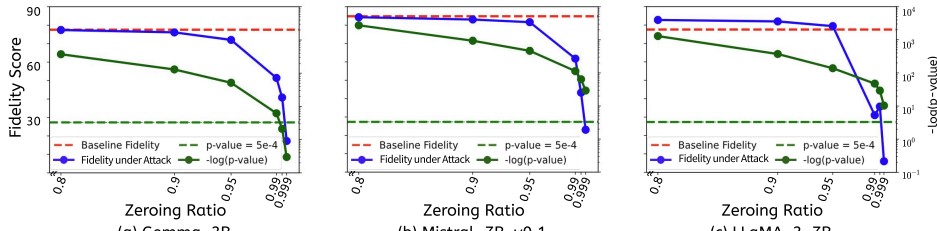

Figure 2: **Pruning attack.** X-axis: zeroing ratio of the smallest parameters in the public adapter $(B', A')$ (L1). Left Y-axis: task fidelity (commonsense micro-average). Right Y-axis: $-\log(\text{p-value})$ $= -\log_{10} p$ for passport detection. We declare detection at $-\log(\text{p-value}) \geq 3.3$ (two-sided test; $\alpha = 5 \times 10^{-4}$). The watermark remains detectable until at least **99.9%** of parameters are zeroed, at which point task fidelity collapses.

Table 5: Finetuning Attack. The detectability of passport on SEAL across either the same ($C_{3e}$ $\rightarrow C_{1e}$ and $I_{3e} \rightarrow I_{1e}$) or different datasets ($C_{3e} \rightarrow I_{1e}$ and $I_{3e} \rightarrow C_{1e}$). Higher is better: larger -ln(p) means stronger rejection of 'extracted key is unrelated to C', i.e., more confident passport detectability. We declare detection if -log(p) $\geq 3.3$ (i.e. $\alpha = 5 \times 10^{-4}$).

| Tasks | Acc. | MT-Bench | -log(p-value) |
|---|---|---|---|
| $C_{3e}$ | 83.1 | - | - |
| $I_{3e}$ | - | 5.81 | - |
| $I_{3e} \rightarrow C_{1e}$ | 60.2 | 4.94 | 79.85 |
| $C_{3e} \rightarrow I_{1e}$ | 0.24 | 3.56 | 79.87 |
| $C_{3e} \rightarrow C_{1e}$ | 82.9 | - | 1824.9 |
| $I_{3e} \rightarrow I_{1e}$ | - | 3.78 | 5.75 |

**Finetuning.** Starting from a public $(B', A')$ trained for three epochs on Commonsense or Alpaca, we resume standard LoRA for one epoch on the same or the other dataset (e.g., $C_{3e} \rightarrow I_{1e}$, $I_{3e} \rightarrow C_{1e}$). Across all cases, the passport remains detectable with $N \approx 10^5$ and $-\log_{10} p \gg 3.3$ (Table 5), supporting robustness to routine post-hoc fine-tuning.

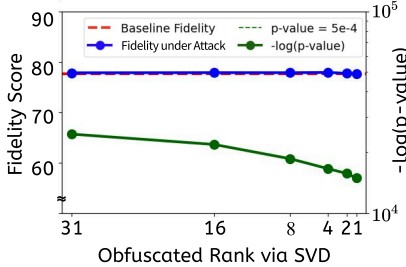

Figure 3: **Structural obfuscation (Gemma-2B via SVD).** Original rank is 32; we obfuscate to ranks $k = 31$ down to 1 via best rank-$k$ projections. Passport detection uses the same two-sided test with $N \approx 10^5$ and the $-\log(\text{p-value}) \geq 3.3$ criterion as in Figure 2.

**Structural obfuscation.** We simulate function-preserving obfuscation by replacing $(B', A')$ with its best rank-$k$ truncated-SVD projection for $k \in \{31, \ldots, 1\}$ from original rank 32 (Yan et al., 2023). As indicated by our spectral diagnostics in Appendix Section I.2, SEAL concentrates energy in early R:d1zQ modes, so the watermark survives until $k$ is very small—fidelity only then drops while extraction stays above the threshold (Figure 3); being function-preserving, these transformations still require passing (**R1**).

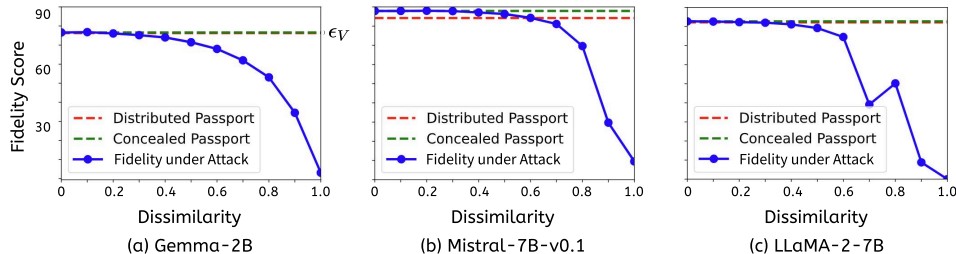

Figure 4: **Ambiguity attacks.** Fidelity $M_T\big(\mathbb{N}(B, A, C_t)\big)$ on commonsense $T$ using an inference-time passport $C_t$ blended as $C_t = (1 - \gamma)C_p + \gamma\, \widetilde{C}_{p\text{-adv}}$ (adversary's matrix). X-axis: dissimilarity $\gamma$. Verification accepts only when the *dual-passport gap* $\Delta_T = \big| M_T(\mathbb{N}(B, A, C)) - M_T(\mathbb{N}(B, A, C_p)) \big|$ is below $\tau_T$ (Table 1); beyond $\gamma \gtrsim 0.6$, the gap typically exceeds $\tau_T$ and claims fail.

**Ambiguity.** Two-passport verification rejects forged keys that were not co-trained (Fan et al., 2019). Table 1 reports owner gaps and thresholds; Figure 4 shows that counterfeit keys must achieve high similarity to the private passport (e.g., $\gamma \gtrsim 0.6$ blending with the true $C_p$) to keep the gap below $\tau_T$, which is implausible without data and co-training. For LLaMA-2-7B and Gemma-2B, owner gaps lie below the Hoeffding bound at the stated $\alpha$; for Mistral-7B the gap exceeds the theoretical bound, so we mark the guarantee as *empirical-only* and list sensitivity.

## 6 CONCLUSION

SEAL is a white-box watermark for LoRA adapters: it inserts non-trainable passports during training and hides them by post-training factorization, so the released adapter is indistinguishable from standard LoRA. We provide an owner-agnostic public verifier (Section 4) that accepts a claim only if it passes **(R0)** structural validity checks, followed by **(R1)** reconstruction and **(R2)** a small dual-passport gap under predeclared thresholds. Across LLM/VLM instruction tuning and text-to-image, SEAL matches LoRA's fidelity while resisting pruning/removal, post-hoc fine-tuning, SVD-style obfuscation, and ambiguity forgeries—non–co-trained keys typically fail the gap test. When the owner's gap satisfies $\widehat{\Delta}_T^{\text{owner}} \leq \tau_T^{\text{theory}}$, we offer a *formal* FPR $\leq \alpha_T$ guarantee; otherwise results are *empirical-only*. The mechanism extends to matmul-style variants and other bilinear operators; we release code and reference thresholds to reproduce the tests and guide task-specific calibration.

## LIMITATIONS

This work targets *adapter-level, white-box* verification for LoRA-style PEFT. The decision rule is statistical and task-dependent: with i.i.d. verifier data and $\widehat{\Delta}_T^{\text{owner}} \leq \tau_T^{\text{theory}}$ we provide a *formal* FPR $\leq \alpha_T$ guarantee; otherwise thresholds are *empirical-only* (Section 4). Fidelity gaps vary by model, task, and rank, so per-task calibration may be needed and our coverage is representative, not exhaustive. Owner-side *extraction* assumes full-rank factors and is intended for owner-in-the-loop checks; recovering $C$ from $(B', A')$ alone is brittle and *not* required by the public verifier.

Regarding the threat of open data, while an adversary may reproduce model behavior by retraining, we clarify that this constitutes *recipe replication* (creating a new artifact) rather than forgery. As demonstrated in our experiments, the specific released artifact remains protected and verifiable even under continued fine-tuning, ensuring that the owner's compute provenance is not erased by downstream modifications.

## REPRODUCIBILITY STATEMENT

We include all artifacts needed to reproduce our results.

1. **Code & configs.** An anonymized repository (linked in the supplementary material) provides training, public verification, and attack scripts, seed-controlled runners, and YAML configs

for every experiment. Upon acceptance we will open-source the repo under a permissive license.

2. **Models & checkpoints.** We rely on official Hugging Face repositories for base models and third-party checkpoints; all use follows their licenses as cited in the Appendix. Our runners fetch these artifacts directly from their sources and reproduce adapters locally from the provided configs and seeds.

3. **Hyperparameters.** Complete settings (ranks, learning rates, batch sizes, optimizers, schedules, epochs) for every model–task pair are listed in the Appendix tables; we also include the exact thresholds used by the verifier.

4. **Evaluation.** Commonsense experiments follow the LLM-Adapters evaluation protocol (**?**). Other tasks use each benchmark's official prompts and scripts; we provide utilities for ROC and $p$-value computation and report $N_T$, $\alpha_T$, and decision criteria (Section 4).

5. **Compute.** GPU types and approximate hours per setting are reported in the Appendix, along with scaled-down recipes to reproduce key figures under limited compute.

All figures and tables can be regenerated via a single entry-point script; required public datasets are downloaded automatically with license checks.

## ETHICS STATEMENT

Watermarking in this paper is *not* cryptography: it provides statistical evidence of ownership (via public tests with stated false-positive control) rather than secrecy or hardness guarantees. Publishing the scheme may aid adversaries; we mitigate by fixing a white-box threat model, using a reproducible decision rule, and releasing code without private passports or proprietary data. Third-party verification presumes either a trusted verifier or an auditable commit-and-reveal of passport hashes recorded at training time; raw keys are revealed only if a dispute arises and must match the commitment. Crucially, our method does not address bias, safety, or legal ownership by itself; the protocol is a reproducible test on parameters—not a legal determination—and should be combined with appropriate licenses and operational controls. All experiments use public datasets under their licenses and involve no human subjects or sensitive data.

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

## A USE OF LARGE LANGUAGE MODELS (LLMS)

We used LLMs only as general-purpose assist tools:

1. **Writing aid.** Grammar/style checking, clarity edits, and minor LaTeX fixes; drafting boilerplate for tables/figures.

2. **Engineering aid.** Boilerplate code for data loaders, evaluation runners, and plotting scripts; all outputs were reviewed and tested by the authors.

3. **Explicit non-usage.** LLMs were *not* used to design the method or ideas, to plan/run experiments or tune hyperparameters, or to produce/alter quantitative results.

4. **Accountability.** All generated content was verified by the authors; prompts and model names are listed in the anonymized code package.

5. **No hidden instructions.** We do not embed hidden instructions, prompts, canary text, or prompt-injection content in the paper, appendix, or supplementary materials; all guidance for reviewers and tools is presented visibly.

## B NOTATION

Table 6: Notation for SEAL. Key symbols and their definitions.

| Symbol | Description |
|---|---|
| $W \in \mathbb{R}^{b \times a}$ | Pretrained backbone (frozen); LoRA/SEAL apply an adaptation on top. |
| $a, b, r$ | Dimensions; $r \ll \min\{a, b\}$. |
| $B \in \mathbb{R}^{b \times r}, \ A \in \mathbb{R}^{r \times a}$ | LoRA's trainable *up/down* factors. |
| $C, \ C_p \in \mathbb{R}^{r \times r}$ | SEAL passports (fixed, non-trainable). $C$ is folded into the public adapter; $C_p$ remains private for verification. |
| $(C_a, C_b)$ | Passports submitted by a *claimant* during public verification (owner: typically $(C, C_p)$). |
| $C_t$ | Runtime passport used at inference/verification (e.g., $C$ for single-passport inference). |
| $f: \ C \mapsto (C_1, C_2)$ | Deterministic factorization with $C_1 C_2 = C$ (e.g., $f_{\text{svd}}$). Publish $(B', A') = (B C_1, \ C_2 A)$. |
| $B', A'$ | Public LoRA adapter after folding $C$ via $f$; same shapes as $B, A$. |
| $\Delta W$ | Weight offset. Standard LoRA: $\Delta W = BA$; SEAL: $\Delta W = BCA$. |
| $\mathbb{N}(\cdot)$ | Adapter operator. Examples: $\mathbb{N}(B, A)$ (LoRA), $\mathbb{N}(B, A, C_t)$ (SEAL with passport $C_t$). |
| $T$ | Host task (e.g., instruction following, QA). |
| $M_T(\cdot)$ | Task metric (e.g., accuracy) used for verification and reporting. |
| $N_T$ | Number of i.i.d. items for $M_T$ (used in theoretical cutoff). |
| $\Delta_T$ | Dual-passport gap: $\big| M_T(\mathbb{N}(B, A, C_a)) - M_T(\mathbb{N}(B, A, C_b)) \big|$. |
| $\rho_T$ | Reconstruction tolerance for (R1): $\|B C_i A - B' A'\|_F \leq \rho_T$. |
| $\delta_C, \kappa_0$ | Policy constants for (R0). $\delta_C$: minimum passport separation threshold; $\kappa_0$: conditioning lower bound derived from factor singular values. R:d1zQ |
| $\tau_T, \ \tau_T^{\text{theory}}$ | (R2) gap cutoff; theoretical bound from Hoeffding and the operational threshold used in practice. |
| $\alpha_T$ | Target false-positive rate for accuracy-type metrics (used to set $\tau_T^{\text{theory}}$). R:d1zQ |
| $-\log_{10} p$ | Detection statistic for extraction tests; we declare detection at $-\log_{10} p \geq -\log_{10} \alpha$ (e.g., 3.3 for $\alpha = 5 \times 10^{-4}$). |
| $\widetilde{B}, \widetilde{A}, \widetilde{C}; \ \widetilde{C}_{p\text{-adv}}$ | Adversarial refactorization of $(B', A')$ and a forged passport used in ambiguity attacks. |

Table 7: Qualitative Comparison with Existing DNN Watermarking Methods. Unlike prior approaches that often introduce additional trainable layers and explicit regularization losses, our method (SEAL) natively integrates into LoRA without extra overhead. BN = batch normalization and GN = group normalization. ♣: Test error, ★: Classification accuracy on AlexNet with CIFAR-100, ♦: FID score. ♠: Accuracy on commonsense reasoning tasks.

| Method | Uchida et al. | Fan et al. | Feng et al. | **SEAL** (Ours) |
|---|---|---|---|---|
| Target Architecture | Convolutional Layer | Normalization Layer | U-Net | LoRA |
| Training Overhead | Regularizer | Regularizer | Latent watermark | Constant matrix |
| Inference Overhead | None | +BN/GN layer | +Secret Enc./Dec. | None |
| Extra Loss Required? | Yes | Yes | Yes | **No** |
| Performance Drop | $\Delta \approx 0.5\%$♣ | $\Delta \approx 1.5\%$★ | $\Delta \approx 2.6\%$♦ | $\Delta \approx 0\%$♠ |
| Attack Resistance | Pruning Finetune | Pruning / Finetune Ambiguity | Pruning Finetune | Pruning / Finetune Ambiguity / Obfuscation |

## C    COMPARISON WITH OTHER DNN WATERMARKING SCHEME

Table 7 qualitatively contrasts four representative DNN watermarking approaches (Uchida et al., 2017; Fan et al., 2019; Feng et al., 2024), and our proposed SEAL. We compare them across multiple dimensions: the targeted network layer, overhead at training/inference time, whether additional loss terms are required, the typical performance drop, and the supported attack resistances. We briefly summarize each row below:

- **Target Architecture.** Each scheme embeds watermarks or passports into different architecture components: convolution layers (Uchida et al., 2017), normalization layers (Fan et al., 2019), U-Net blocks (Feng et al., 2024), and LoRA blocks (SEAL, ours). Our approach focuses on LoRA, a lightweight adapter mechanism.

- **Training Overhead.** Methods like (Uchida et al., 2017; Fan et al., 2019) use an explicit regularizer to embed watermarks, while (Feng et al., 2024) attaches latent-watermark modules during training. In contrast, SEAL entangles a constant matrix with LoRA's low-rank modules, introducing minimal overhead at training time.

- **Inference Overhead.** Despite some methods adding new layers or requiring a secret encoder/decoder at inference, SEAL has no additional components during inference. Once merged, our constant matrix seamlessly integrates into the LoRA parameters.

- **Extra Loss Required?** Most existing watermarking approaches rely on an additional loss term for embedding or regularizing. Our scheme needs *no* extra loss, as the constant matrix naturally entangles with LoRA blocks during the normal training objective.

- **Performance Drop.** We list the reported performance degradation $\Delta$ under each approach, measured by various metrics: (♣) test error, (★) classification accuracy drop, (♦) FID score changes, and (♠) commonsense reasoning tasks. Our SEAL achieves near-zero ($\Delta \approx 0\%$) degradation.

- **Attack Resistance.** We indicate which attacks each method defends against attacks (e.g. pruning, fine-tuning, ambiguity, or obfuscation attack). Our SEAL covers a broader range of threats in a white-box setting, including pruning, fine-tuning, obfuscation, and ambiguity.

Our approach stands out for its simpler training pipeline (*no* explicit regularizer), near zero inference overhead, and broader attack coverage, all while incurring practically zero performance drop.

# D  TRAINING PROCESS OF SEAL

## D.1  FORWARD PATH

In SEAL, the forward path produces the output $W'$ by adding a learnable offset $\Delta W$ on top of the base weights $W$:

$$W' = W + \Delta W = W + BCA. \tag{7}$$

Here, $B$ and $A$ are trainable matrices, while $C$ is a fixed *passport* matrix that carries the watermark. Unlike traditional LoRA layers that use $\Delta W = BA$ alone, SEAL inserts $C$ between $B$ and $A$. This additional matrix:

- Forces the resulting offset $\Delta W$ to pass through an extra linear transformation, potentially mixing or reorienting the learned directions.
- Ties the final weight update $\Delta W$ to the presence of $C$; removing or altering $C$ would disrupt $\Delta W$ and hence the model's functionality.

If $C$ were diagonal, it would merely scale each dimension independently, which can be easier to isolate or undo. However, when $C$ is a full (non-diagonal) matrix, the learned offset $\Delta W$ may exhibit more complex structures, as the multiplication by $C$ intermixes channels or dimensions.

## D.2  BACKWARD PATH

The backward path computes gradients of the loss function $\phi$ with respect to $A$ and $B$, revealing how $C$ influences the updates. Let

$$\Delta := BCA \quad \text{and} \quad \Phi := \phi(\Delta x), \tag{8}$$

where $\Delta x$ represents applying $\Delta$ to some input $x$. Then, by the chain rule,

$$\frac{\partial \Phi}{\partial A} = (BC)^T \frac{\partial \phi}{\partial \Delta} = C^T B^T \frac{\partial \phi}{\partial \Delta}, \tag{9}$$

$$\frac{\partial \Phi}{\partial B} = \frac{\partial \phi}{\partial \Delta} (CA)^T = \frac{\partial \phi}{\partial \Delta} A^T C^T. \tag{10}$$

These expressions highlight two key points:

**(1) Transformation of Gradients.** Each gradient, $\nabla_A$ and $\nabla_B$, is multiplied (from the left or right) by $C^T$. If $C$ were diagonal, this would reduce to element-wise scaling of the gradient, which is relatively simple to reverse or interpret. In contrast, a *full* $C$ applies a more general linear transformation—potentially a rotation or mixing—to the gradient directions.

**(2) Entanglement of Learnable Parameters.** Because $C$ is *fixed* but non-trivial, both $B$ and $A$ are continually updated in a manner dependent on $C$. Over many gradient steps, $\Delta W = BCA$ becomes *entangled* across multiple dimensions; single-direction modifications in $B$ or $A$ cannot easily isolate the watermark without affecting other directions.

## D.3  TRAINING EFFICIENCY ANALYSIS

To quantify the cost of entanglement, we compare the training time and peak memory usage of SEAL against Standard LoRA and DoRA (Liu et al., 2024b) on the LLaMA-2-7B commonsense reasoning task (4×RTX 3090, global batch size 16). As shown in Table 8, SEAL incurs a moderate time overhead (∼63%) due to the dual-passport forward passes, which is comparable to other advanced adapters like DoRA. Crucially, the memory overhead is marginal (∼2%), making SEAL practical for memory-constrained environments. R:fNRL, WQP7

Table 8: **Training Overhead Comparison.** Measured on LLaMA-2-7B (Commonsense) for 3 epochs. SEAL adds security with manageable cost, comparable to DoRA.

| Method | Time (h) | Rel. Time | Memory (GB) | Rel. Mem |
|---|---|---|---|---|
| LoRA | 12.0 | 1.00× | 16.80 | 1.00× |
| DoRA | 18.5 | 1.54× | - | - |
| **SEAL (Ours)** | **19.6** | **1.63×** | **17.13** | **1.02×** |

# E   THEORETICAL ANALYSIS: CONDITIONING, UNIQUENESS, AND FORGERY RESISTANCE

This section provides the mathematical foundation for our verification policy **(R0)-(R2)**. We first derive the conditioning lower bound (Lemma 1) that enforces structural separation, proving that distinct passports induce distinct weight updates. We then validate this bound empirically and finally discuss why this theoretical constraint makes post-hoc forgery computationally infeasible. R:d1zQ, P2Pv

## E.1   PROOF OF CONDITIONING LOWER BOUND (LEMMA 1)

**Lemma 1.** *Let $\Delta W(C) = BCA$ be the adapter update parameterized by passport $C$. If $B$ and $A$ are full-rank matrices with singular values $\sigma_i(B)$ and $\sigma_j(A)$, then for any two passports $C_1, C_2$, the Frobenius norm of the induced weight difference is lower-bounded by:*

$$\|\Delta W(C_1) - \Delta W(C_2)\|_F \geq \kappa_0 \|C_1 - C_2\|_F, \tag{11}$$

*where $\kappa_0 = \min_{i,j}(\sigma_i(B) \cdot \sigma_j(A)) > 0$.*

**Proof.**   Let $X = C_1 - C_2$. We rely on the relationship between the Frobenius norm of a matrix and the Euclidean norm of its vectorization: $\|M\|_F = \|\text{vec}(M)\|_2$. The difference in weight space is linear in $X$:

$$\Delta W(C_1) - \Delta W(C_2) = B(C_1 - C_2)A = BXA. \tag{12}$$

Applying the vectorization operator and the identity $\text{vec}(BXA) = (A^\top \otimes B)\text{vec}(X)$, where $\otimes$ is the Kronecker product:

$$\text{vec}(BXA) = (A^\top \otimes B)\mathbf{x}, \quad \text{where } \mathbf{x} = \text{vec}(X). \tag{13}$$

The singular values of the Kronecker product $A^\top \otimes B$ are exactly the pairwise products of singular values of $A$ and $B$, i.e., $\{\sigma_j(A)\sigma_i(B)\}_{i,j}$. Let $\sigma_{\min}(M)$ denote the smallest singular value of a matrix $M$. For any vector $\mathbf{v}$, $\|M\mathbf{v}\|_2 \geq \sigma_{\min}(M)\|\mathbf{v}\|_2$. Therefore,

$$\|\text{vec}(BXA)\|_2 \geq \min_{i,j}(\sigma_j(A)\sigma_i(B)) \cdot \|\text{vec}(X)\|_2 \tag{14}$$

$$\|BXA\|_F \geq \kappa_0 \|X\|_F, \tag{15}$$

where $\kappa_0 = \sigma_{\min}(B)\sigma_{\min}(A)$. Since $(B', A')$ are released as standard LoRA weights (typically initialized and trained to be full-rank), $\kappa_0$ is strictly positive, ensuring that the mapping from passport difference to weight difference is a bijection (and isomorphism up to scaling) on the passport subspace. ☐ R:d1zQ, P2Pv, WQP7

## E.2   UNIQUENESS OF THE PASSPORT UNDER FULL-RANK FACTORS

**Assumptions.**   The attacker fixes rank-$r$ matrices $\widetilde{B} \in \mathbb{R}^{b \times r}$ and $\widetilde{A} \in \mathbb{R}^{r \times a}$ with $\text{rank}(\widetilde{B}) = \text{rank}(\widetilde{A}) = r$.

**Claim (full-rank uniqueness).**   If two passports $\widetilde{C}, \widetilde{C}_{p\text{-adv}} \in \mathbb{R}^{r \times r}$ satisfy $\widetilde{B}\,\widetilde{C}\,\widetilde{A} = \widetilde{B}\,\widetilde{C}_{p\text{-adv}}\,\widetilde{A} = B'A'$, then $\widetilde{C} = \widetilde{C}_{p\text{-adv}}$.

**Proof.**   Since $\widetilde{B}$ has full column rank, there exists a left inverse $L \in \mathbb{R}^{r \times b}$ with $L\widetilde{B} = I_r$. Since $\widetilde{A}$ has full row rank, there exists a right inverse $R \in \mathbb{R}^{a \times r}$ with $\widetilde{A}R = I_r$. Subtracting the two equalities and multiplying on the left/right gives

$$L\,\widetilde{B}\,(\widetilde{C} - \widetilde{C}_{p\text{-adv}})\,\widetilde{A}\,R = I_r\,(\widetilde{C} - \widetilde{C}_{p\text{-adv}})\,I_r = \widetilde{C} - \widetilde{C}_{p\text{-adv}} = \mathbf{0}_{r \times r}.$$

Hence $\widetilde{C} = \widetilde{C}_{p\text{-adv}}$. ☐

**Remark on Rank Deficiency.**   If factors are rank-deficient ($< r$), uniqueness holds only up to the null space. However, as discussed in Lemma 1 and the empirical section below, trained adapters maintain full rank and robust conditioning, making this theoretical degeneracy irrelevant for practical verification.

### E.3 Empirical Validation of Conditioning ($\kappa_0$)

To validate that the theoretical lower bound $\kappa_0$ is non-trivial in practice, we measured the conditioning constant across multiple training runs on Gemma-2B ($r = 32$). As shown in Table 9, $\kappa_0$ remains consistently positive across different seeds and passport types. This confirms that the linear map from passport to weight update is well-conditioned, effectively enforcing the separation constraint required by Policy (R0).

R:d1zQ, P2Pv

Table 9: **Empirical validation of Conditioning Constant ($\kappa_0$).** Measured on Gemma-2B ($r = 32$) across 3 random seeds. $\kappa_{\min}$ represents the global lower bound used for the R0 policy check. The values are consistently positive, confirming non-degeneracy.

| Seed | Key Variant | Layers | $\kappa_{\min}$ | $\kappa_{\max}$ | $\kappa_{\mathbf{mean}}$ | $\kappa_{\mathbf{std}}$ |
|------|-------------|--------|-----------------|-----------------|--------------------------|-------------------------|
| #1 | Bitmap ($C$) | 90 | $1.33 \times 10^{-6}$ | $4.04 \times 10^{-5}$ | $1.05 \times 10^{-5}$ | $1.02 \times 10^{-5}$ |
| | Random ($C_p$) | 90 | $2.44 \times 10^{-6}$ | $2.02 \times 10^{-3}$ | $3.65 \times 10^{-4}$ | $4.31 \times 10^{-4}$ |
| #2 | Bitmap ($C$) | 90 | $7.18 \times 10^{-7}$ | $4.53 \times 10^{-5}$ | $1.06 \times 10^{-5}$ | $1.07 \times 10^{-5}$ |
| | Random ($C_p$) | 90 | $2.70 \times 10^{-6}$ | $2.07 \times 10^{-3}$ | $3.80 \times 10^{-4}$ | $4.53 \times 10^{-4}$ |
| #3 | Bitmap ($C$) | 90 | $1.56 \times 10^{-6}$ | $5.11 \times 10^{-5}$ | $1.07 \times 10^{-5}$ | $1.06 \times 10^{-5}$ |
| | Random ($C_p$) | 90 | $2.28 \times 10^{-6}$ | $2.51 \times 10^{-3}$ | $3.91 \times 10^{-4}$ | $4.85 \times 10^{-4}$ |

### E.4 Infeasibility of Forgery: Connecting Theory to Defense

Based on the theoretical and empirical results above, we clarify why forging a valid passport pair is computationally infeasible. The defense relies on the **concurrent entanglement** of $B$, $A$ with both $C$ and $C_p$ during training, which post-hoc factorization cannot replicate.

1. **Tension between R0 and R2 (via Lemma 1).** To pass the public verification, an adversary must submit a passport $C_{p\text{-adv}}$ that is structurally distinct from $C$ (satisfying R0: $\|C - C_{p\text{-adv}}\|_F > \delta$) but functionally identical (satisfying R2: gap $\approx 0$). Lemma 1 dictates that structural distance induces weight distance: $\|\Delta W(C) - \Delta W(C_{p\text{-adv}})\|_F \geq \kappa_0 \delta$. Since the adversary's weights $(\widetilde{B}, \widetilde{A})$ were not co-trained to map these distinct weights to the same functional output, satisfying R2 becomes a hard inverse problem.

2. **Computational Cost.** Solving this inverse problem requires optimizing $C_{p\text{-adv}}$ to minimize the validation metric gap $\Delta_T$. This is far more expensive than standard training because it requires two full inference passes per optimization step to evaluate $\Delta_T$, with no gradient access to the non-differentiable metric $M_T$.

3. **Indistinguishability** Furthermore, as detailed in Section 1, the released weights $(B', A')$ are structurally identical to standard LoRA. Without insider knowledge, an attacker cannot even distinguish a SEAL-protected adapter from a standard one.

4. **Limited Payoff** Even if the training data is public, our fine-tuning experiments confirm that the watermark remains detectable in the original artifact, as shown in 5.4. Consequently, an adversary capable of retraining gains no practical advantage by attacking the specific released weights; since they possess the data and compute to train a legitimate model, attempting such a costly forgery yields no economic utility and implies purely malicious intent without profit.

R:d1zQ, P2Pv, WQP7

## F   EXTENSIONS TO MATMUL-BASED LoRA VARIANTS

Beyond the canonical LoRA (Hu et al., 2022) formulation, numerous follow-up works propose modifications and enhancements while still employing matrix multiplication (`matmul`) as the underlying low-rank adaptation operator. In this section, we illustrate how SEAL is compatible or can be adapted to these matmul-based variants. Although we do not exhaustively enumerate every LoRA-derived approach, the general principle remains: if the adaptation primarily uses matrix multiplication (possibly with additional diagonal, scaling, or regularization terms), then SEAL can often be inserted by embedding a non-trainable passport $C$ between the *up* and *down* blocks.

### F.1   LoRA-FA (ZHANG ET AL., 2023A)

LoRA-FA (LoRA with frozen down blocks) modifies LoRA by keeping the *down* block frozen during training, while only the *up* block is trained. Structurally, however, it does not alter the fundamental `matmul` operator. Consequently, integrating SEAL follows the same procedure as standard LoRA: one can embed the passport $C$ into the product $B\,C\,A$ without requiring any special adjustments. The difference in training rules (i.e. freezing $A$) does not affect how $C$ is placed or how it is decomposed into $(C_1, C_2)$ for final public release.

### F.2   LoRA+ (HAYOU ET AL., 2024)

LoRA+ investigates the training dynamics of LoRA's *up* ($B$) and *down* ($A$) blocks. In particular, it emphasizes the disparity in gradient magnitudes and proposes using different learning rates:
$$A \;\leftarrow\; A \,-\, \eta\,G_A, \quad B \;\leftarrow\; B \,-\, \lambda\eta\,G_B,$$
where $\lambda \gg 1$ is a scale factor, $\eta$ is the base learning rate, and $G_A, G_B$ are the respective gradients. LoRA+ does *not* alter the structural operator (still matrix multiplication). Therefore, SEAL can be employed by introducing $C \in \mathbb{R}^{r \times r}$ between $B$ and $A$, yielding $\Delta W = B\,C\,A$. The difference in gradient scaling does not impact the usage of a non-trainable passport matrix $C$.

### F.3   VeRA (KOPICZKO ET AL., 2024)

VeRA introduces two diagonal matrices, $\Lambda_b$ and $\Lambda_d$, to scale different parts of the low-rank factors:
$$\Delta W \;=\; \Lambda_b\,B\,\Lambda_d\,A,$$
where $B, A$ may be random, frozen, shared across layers and the diagonal elements in $\Lambda_b, \Lambda_d$ are trainable. Despite these diagonal scalings, the core operator remains matrix multiplication. Hence, embedding a passport $C$ is still feasible. By leveraging the commutative property of diagonal matrices and $C$ (assuming $C$ commutes with $\Lambda_d$ in the sense that one can re-factor $C$ into $C_1\Lambda_d C_2$ or $\Lambda_d C$), SEAL can be inserted:

$$\Delta W \;=\; \Lambda_b\,(B\,C_1)\,\Lambda_d\,(C_2\,A),$$
which is functionally identical to $\Lambda_b\,B\,\Lambda_d\,A$ except for the hidden passport $C = C_1 C_2$. Implementing SEAL in VeRA may require converting the final trained weights back into a standard $(B', A')$ form plus a diagonal scaling term, but the fundamental principle is straightforward.

### F.4   ADALoRA (ZHANG ET AL., 2023B)

AdaLoRA applies a dynamic rank-allocating approach inspired by SVD. It factorizes the weight update into:
$$\Delta W \;=\; P\,\Lambda\,Q,$$
where $\Lambda$ is a diagonal matrix, and $P, Q$ are regularized to maintain near-orthogonality. Since diagonal matrices commute under multiplication (up to a re-factorization), one can embed a passport $C$ by decomposing it ($f(C) \to (C_1, C_2)$). In essence,
$$\Delta W \;=\; P\,C_1\,\Lambda\,C_2\,Q \;=\; P'\,\Lambda\,Q',$$
where $P' = PC_1$ and $Q' = C_2 Q$. This preserves the rank-$r$ structure and does not disrupt AdaLoRA's optimization logic. Regularization terms that enforce $P'^T P' \approx I$ and $Q'Q'^T \approx I$ remain valid, though one may incorporate $C_1, C_2$ into the initialization or adapt them carefully so as not to degrade the orthogonality constraints.

## F.5   DoRA (Liu et al., 2024b)

DoRA modifies the final LoRA update using a column-wise norm factor:

$$W' \;=\; \frac{\|W\|_c}{\|\,W + \Delta W\,\|_c}\,(W + \Delta W),$$

where $\|\cdot\|_c$ computes column-wise norms and the ratio is (by design) often detached from gradients to reduce memory overhead. Replacing $\Delta W$ with $B\,C\,A$ in DoRA does not alter the external gradient manipulation logic, since $C$ is non-trainable. Thus,

$$W' \;=\; \frac{\|W\|_c}{\|\,W + B\,C\,A\,\|_c}\,(W + B\,C\,A)$$

remains valid. The presence of $C$ does not interfere with DoRA's approach to scaling or norm-based constraints.

## F.6   Integrating with DoRA

Table 10: Commonsense Reasoning on Llama-2-7B for LoRA, DoRA, SEAL. SEAL+DoRA is a combined approach. Hyperparameters in Table 18.

| Method | Wall Time (h) | Avg. |
|---|---|---|
| LoRA | 12.0 | $81.67_{\pm 1.03}$ |
| DoRA | 18.5 | $81.98_{\pm 0.26}$ |
| SEAL | 19.6 | $\mathbf{83.78}_{\pm 0.27}$ |
| SEAL + DoRA | 27.8 | $81.88_{\pm 1.08}$ |

Thanks to its flexible framework, SEAL can easily be applied to a wide variety of LoRA variants. In Table 10, we use DoRA (Liu et al., 2024b) as a case study to demonstrate that SEAL can seamlessly integrate with diverse LoRA-based methods, as exemplified by SEAL+DoRA. We measure wall time on four RTX 3090 GPUs. DoRA requires magnitude and direction computations, while SEAL's passport training also adds overhead. Still, SEAL+DoRA achieves near-DoRA accuracy.

## F.7   Variants with Non-Multiplicative Operations

All of the above variants preserve the core LoRA assumption of a matrix multiplication operator for the rank-$r$ adaptation. However, certain approaches introduce non-multiplicative adaptations (e.g., Hadamard product, Kronecker product, or other specialized transforms). In the following section, for these cases, which discuss how SEAL can be generalized to any bilinear or multilinear operator $\star$.

# G   Extensions to Generalized Low-Rank Operators

In the main text, we considered a standard LoRA (Hu et al., 2022) that uses a matrix multiplication operator:

$$\Delta W = B\,C\,A,$$

where $B \in \mathbb{R}^{b \times r}$, $C \in \mathbb{R}^{r \times r}$, and $A \in \mathbb{R}^{r \times a}$. Recent work has explored alternative low-rank adaptation mechanisms beyond simple `matmul`, such as Kronecker product-based methods (Edalati et al., 2022; Yeh et al., 2023) or even elementwise (Hadamard) product (Hyeon-Woo et al., 2021) forms. Our approach can be extended in a straightforward manner to these generalized operators, which we denote as $\star$.

## G.1   General Operator $\star$

Let $\star$ be any bilinear or multilinear operator used for low-rank adaptation.[1] We can then write the trainable adaptation layer as

$$\Delta W = B \,\star\, C \,\star\, A,$$

---

[1]Here, *bilinear* means $(X \star Y)$ is linear in both $X$ and $Y$ when one is held fixed, e.g. standard matrix multiplication, Kronecker product, or Hadamard product.

where $B, A$ are the trainable low-rank parameters, and $C$ is the non-trainable passport in SEAL. During training, $B$ and $A$ are optimized in conjunction with $C$ held fixed (just as in the matrix multiplication case).

**Decomposition Function for Operator $\star$.** To *distribute* $C$ into $(B, A)$ after training, we require a *decomposition function* $f : C \mapsto (C_1, C_2)$ such that

$$C = C_1 \, \star \, C_2.$$

For example, under the Kronecker product $\otimes$, one could define $f(C)$ to split $C$ into smaller block partitions, or use an SVD-like factorization in an appropriate transformed space. Under the Hadamard product, $f(C)$ could involve elementwise roots or other transformations.

Once $C_1$ and $C_2$ are obtained, we apply:

$$B' \, = \, B \, \star \, C_1 \quad , \quad A' \, = \, C_2 \, \star \, A,$$

so that

$$B' \, \star \, A' \; = \; (B \, \star \, C_1) \star (C_2 \star A) \; = \; B \, \star \, (C_1 \star C_2) \, \star \, A \; = \; B \, \star \, C \, \star \, A.$$

Hence, the final distributed weights $(B', A')$ for public remain *functionally equivalent* to using $B, A, C$.

### G.2 IMPLICATIONS AND FUTURE DIRECTIONS

- **Broader Applicability.** By permitting $\star$ to be any bilinear or multilinear operator (Kronecker, Hadamard, etc.), SEAL naturally extends beyond the canonical matrix multiplication used in most LoRA implementations. This flexibility can be valuable for advanced parameter-efficient tuning methods (Edalati et al., 2022; Hyeon-Woo et al., 2021; Yeh et al., 2023).

- **Same Security Guarantees.** The central watermarking principle (embedding a non-trainable passport $C$ into the adaptation) does not change. An adversary attempting to re-factor $B' \star A'$ to recover $C$ faces the same challenges described in the main text and Appendix **??**—non-identifiability, cost of reconstruction, and multi-passport verification barriers.

- **Potential Operator-Specific Designs.** Certain operators (e.g., Kronecker product) may admit additional constraints or factorization strategies that could be exploited for improved stealth or efficiency. Investigating these is an interesting direction for future work.

In summary, SEAL can be generalized to other operators $\star$ by treating $C$ as a non-trainable factor and defining a suitable decomposition function $f(C)$ such that $C = C_1 \star C_2$. This allows us to hide the passport just as in the matrix multiplication case, thereby preserving the main SEAL pipeline for more complex LoRA variants.

Table 11: Hyperparameter configurations of SEAL and LoRA for Gemma-2B, Mistral-7B-v0.1, LLaMA2-7B/13B, and LLaMA3-8B on the commonsense reasoning. All experiments are done with 4x A100 80GB (for LLaMA-2-13B) and 4x RTX 3090 (for the other models) with approximately 15 hours.

| Models | Gemma-2B | | Mistral-7B-v0.1 | | LLaMA-2-7B | | LLaMA-2-13B | | LLaMA-3-8B | |
|---|---|---|---|---|---|---|---|---|---|---|
| Method | LoRA | SEAL | LoRA | SEAL | LoRA | SEAL | LoRA | SEAL | LoRA | SEAL |
| r | | | | | 32 | | | | | |
| alpha | | | | | 32 | | | | | |
| Dropout | | | | | 0.05 | | | | | |
| LR | 2e-4 | 2e-5 | 2e-5 | 2e-5 | 2e-4 | 2e-5 | 2e-4 | 2e-5 | 2e-4 | 2e-5 |
| Optimizer | | | | AdamW Loshchilov & Hutter (2019) | | | | | | |
| LR scheduler | | | | | Linear | | | | | |
| Weight Decay | | | | | 0 | | | | | |
| Warmup Steps | | | | | 100 | | | | | |
| Total Batch size | | | | | 16 | | | | | |
| Epoch | | | | | 3 | | | | | |
| Target Modules | | | | Query Key Value UpProj DownProj | | | | | | |

Table 12: Hyperparameter configurations of SEAL and LoRA for Instruction Tuning. All experiments are done with 1x A100 80GB for approximately 2 hours. All w/o LM HEAD are Query, Key, Value, Out, UpProj, DownProj, GateProj.

| Model | LLaMA-2-7B | |
|---|---|---|
| Method | LoRA | SEAL |
| r | 32 | |
| alpha | 32 | |
| Dropout | 0.0 | |
| LR | 2e-5 | |
| LR scheduler | Cosine | |
| Optimizer | AdamW | |
| Weight Decay | 0 | |
| Total Batch size | 8 | |
| Epoch | 3 | |
| Target Modules | All w/o LM HEAD | |

## H TRAINING DETAILS

### H.1 COMMONSENSE REASONING TASKS

The hyperparameters used for these evaluations are listed in Table 18.

### H.2 TEXTUAL INSTRUCTION TUNING

We conducted textual instruction tuning using Alpaca dataset (Taori et al., 2023) on LLaMA-2-7B (Touvron et al., 2023), trained for 3 epochs. The hyperparameters used for this process are detailed in Table 12.

### H.3 VISUAL INSTRUCTION TUNING

We compared the fidelity of SEAL, LoRA, and FT on the visual instruction tuning tasks with LLaVA-1.5-7B (Liu et al., 2024a). To ensure a fair comparison, we used the same original model provided by (Liu et al., 2024a) uses the same configuration as the LoRA setup with the same training dataset. We adhere to (Liu et al., 2024a) setting to filter the training data and design the tuning prompt format.

Table 13: Performance comparison of different methods across seven visual instruction tuning benchmarks

| Method | # Params (%) | VQAv2 | GQA | VisWiz | SQA | VQAT | POPE | MMBench | Avg |
|--------|-------------|-------|-----|--------|-----|------|------|---------|-----|
| FT | 100 | 78.5 | 61.9 | 50.0 | 66.8 | 58.2 | 85.9 | 64.3 | 66.5 |
| LoRA | 4.61 | 79.1 | 62.9 | 47.8 | 68.4 | 58.2 | 86.4 | 66.1 | **66.9** |
| SEAL | 4.61 | 75.4 | 58.3 | 41.6 | 66.9 | 52.9 | 86.0 | 60.5 | 63.1 |

Table 14: Hyperparameters for visual instruction tuning. All experiments were performed with 4x A100 80GB with approximately 24 hours.

| Model | LLaVA-1.5-7B | |
|-------|------|------|
| Method | LoRA | SEAL |
| r | 128 | |
| alpha | 128 | |
| LR | 2e-4 | 2e-5 |
| LR scheduler | Linear | |
| Optimizer | AdamW | |
| Weight Decay | 0 | |
| Warmup Ratio | 0.03 | |
| Total Batch size | 64 | |

## H.4 TEXT-TO-IMAGE SYNTHESIS

The DreamBooth dataset (Ruiz et al., 2023) encompasses 30 distinct subjects from 15 different classes, featuring a diverse array of unique objects and live subjects, including items such as backpacks and vases, as well as pets like cats and dogs. Each of the subjects contains 4-6 images. These subjects are categorized into two primary groups: inanimate objects and live subjects/pets. Of the 30 subjects, 21 are dedicated to objects, while the remaining 9 represent live subjects/pets.

For subject fidelity, following (Ruiz et al., 2023), we use CLIP-I, DINO. CLIP-I, an image-text similarity metric, compares the CLIP (Radford et al., 2021) visual features of the generated images with those of the same subject images. DINO (Caron et al., 2021), trained in a self-supervised manner to distinguish different images, is suitable for comparing the visual attributes of the same object generated by models trained with different methods. For prompt fidelity, the image-text similarity metric CLIP-T compares the CLIP features of the generated images and the corresponding text prompts without placeholders, as mentioned in (Ruiz et al., 2023; Nam et al., 2024). For the evaluation, we generated four images for each of the 30 subjects and 25 prompts, resulting in a total of 3,000 images. The prompts used for this evaluation are identical to those originally used in (Ruiz et al., 2023) to ensure consistency and comparability across models. These prompts are designed to evaluate subject fidelity and prompt fidelity across diverse scenarios, as detailed in Table 15.

Figure 5 visually compares LoRA and SEAL on representative subjects from the DreamBooth dataset. The top row shows example reference images for each subject, the middle row shows images generated by LoRA, and the bottom row shows images from our SEAL. Qualitatively, both methods faithfully capture key attributes of each subject (e.g., shape, color, general pose) and produce images of comparable visual quality. That is, SEAL does not degrade or alter the original subject's appearance relative to LoRA, suggesting that incorporating the constant matrix $C$ does not introduce noticeable artifacts or reduce fidelity. These results align with the quantitative metrics on subject and prompt fidelity, indicating that SEAL maintains a quality level on par with LoRA while embedding a watermark in the learned parameters.

Table 15: DreamBooth text prompts used for evaluation of inanimate objects and live subjects.

| Prompts for Non-Live Objects | Prompts for Live Subjects |
|---|---|
| a {} in the jungle | a {} in the jungle |
| a {} in the snow | a {} in the snow |
| a {} on the beach | a {} on the beach |
| a {} on a cobblestone street | a {} on a cobblestone street |
| a {} on top of pink fabric | a {} on top of pink fabric |
| a {} on top of a wooden floor | a {} on top of a wooden floor |
| a {} with a city in the background | a {} with a city in the background |
| a {} with a mountain in the background | a {} with a mountain in the background |
| a {} with a blue house in the background | a {} with a blue house in the background |
| a {} on top of a purple rug in a forest | a {} on top of a purple rug in a forest |
| a {} with a wheat field in the background | a {} wearing a red hat |
| a {} with a tree and autumn leaves in the background | a {} wearing a santa hat |
| a {} with the Eiffel Tower in the background | a {} wearing a rainbow scarf |
| a {} floating on top of water | a {} wearing a black top hat and a monocle |
| a {} floating in an ocean of milk | a {} in a chef outfit |
| a {} on top of green grass with sunflowers around it | a {} in a firefighter outfit |
| a {} on top of a mirror | a {} in a police outfit |
| a {} on top of the sidewalk in a crowded street | a {} wearing pink glasses |
| a {} on top of a dirt road | a {} wearing a yellow shirt |
| a {} on top of a white rug | a {} in a purple wizard outfit |
| a red {} | a red {} |
| a purple {} | a purple {} |
| a shiny {} | a shiny {} |
| a wet {} | a wet {} |
| a cube shaped {} | a cube shaped {} |

Figure 5: Qualitative comparison of LoRA and SEAL in Text-to-Image Synthesis task

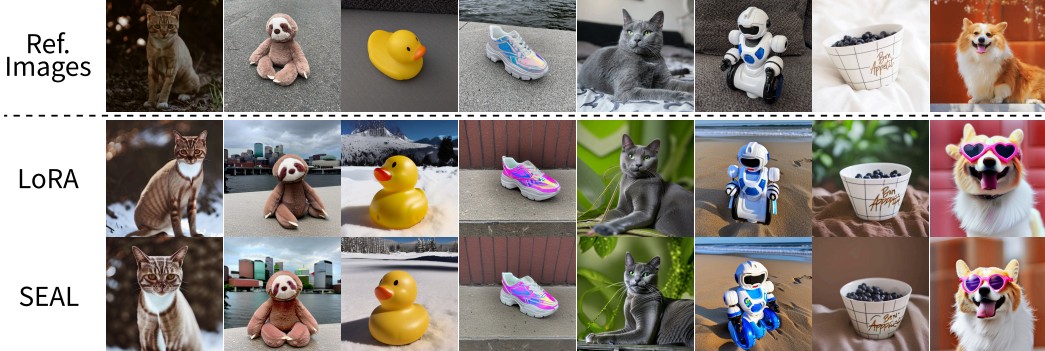

Figure 6: Qualitative comparison of passports in Text-to-Image Synthesis task

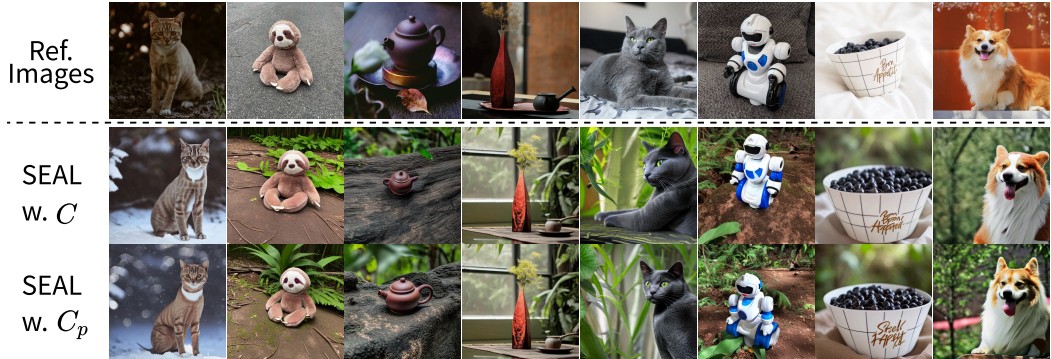

Table 16: Hyperparameter configurations of SEAL and LoRA for Text-to-Image Synthesis. All experiments are done with 4x RTX 4090 with approximately 15 minutes per subject.

| Model | Stable Diffusion 1.5 | |
|---|---|---|
| Method | LoRA | SEAL |
| r | | 32 |
| alpha | | 32 |
| Dropout | | 0.0 |
| LR | 5e-5 | 1e-5 |
| LR scheduler | | Constant |
| Optimizer | | AdamW |
| Weight Decay | | 1e-2 |
| Total Batch size | | 32 |
| Steps | | 300 |
| Target Modules | Q K V Out AddK AddV | |

Table 17: Hyperparameter configurations of Finetruning Attack on SEAL which trains on 3-epoch. We resume training on $\mathbb{N}(B', A')$, which passport $C$ is distributed in $B, A$ via $f_{svd}$.

| Model | LLaMA-2-7B |
|---|---|
| Method | LoRA |
| r | 32 |
| alpha | 32 |
| LR | 2e-5 |
| Optimizer | AdamW |
| LR scheduler | Linear |
| Weight Decay | 0 |
| Warmup Steps | 100 |
| Batch size | 16 |
| Epoch | 1 |
| Target Modules | Query Key Value UpProj DownProj |

Table 18: Hyperparameter configurations of Integrating with DoRA.

| Model | LLaMA-2-7B | | | |
|---|---|---|---|---|
| Method | LoRA | SEAL | DoRA | SEAL+DoRA |
| r | | | 32 | |
| alpha | | | 32 | |
| Dropout | | | 0.05 | |
| LR | 2e-4 | 2e-5 | 2e-4 | 2e-5 |
| Optimizer | | | AdamW | |
| LR scheduler | | | Linear | |
| Weight Decay | | | 0 | |
| Warmup Steps | | | 100 | |
| Total Batch size | | | 16 | |
| Epoch | | | 3 | |
| Target Modules | | Query Key Value UpProj DownProj | | |

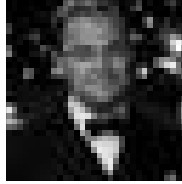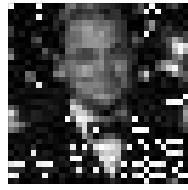

Figure 7: **Passport Example**. *Left*: A 32×32 grayscale bitmap (cropped and downsampled from a YouTube clip[2]) serves as our non-trainable passport $C$. *Right*: The passport partially recovered (from 10% zeroed SEAL weight on LLaMA-2-7B).

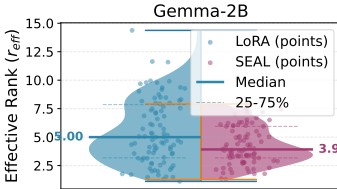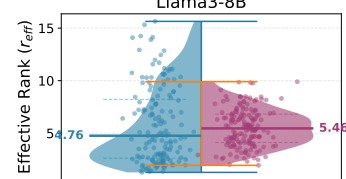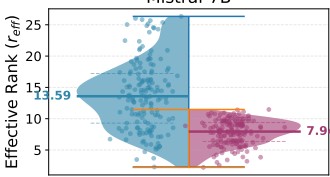

Figure 8: **Effective-rank distributions** of $\Delta W$ across layers. $r_{\text{eff}} = \exp(-\sum_i p_i \log p_i)$ with $p_i = \sigma_i^2 / \sum_j \sigma_j^2$. Split violins: LoRA (left), SEAL (right).

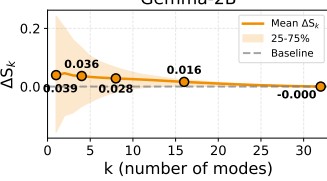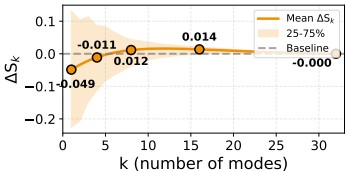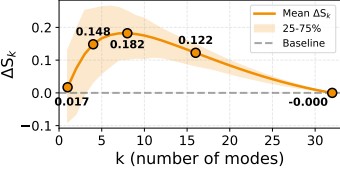

Figure 9: **Cumulative-energy difference** $\Delta S_k$ (SEAL−LoRA) across backbones. $S_k = \sum_{i=1}^{k} p_i$; curves show layer-wise mean with IQR bands. Positive values at small $k$ indicate more top-mode concentration under SEAL.

# I   ABLATION STUDY

## I.1   PASSPORT EXAMPLE

In order to provide a concrete illustration of our watermark extraction process, we construct a small 32×32 grayscale image as the *passport* $C$ (or $C_p$). Specifically, we sampled 100 frames from a publicly available YouTube clip, applied center-cropping on each frame, converted them to grayscale, and then downsampled to 32×32. From these frames, we selected one representative image (shown in Figure 7) to embed as the non-trainable matrix $C$ in our SEAL pipeline Section 3.

This tiny passport image, while derived from a movie clip, is both *unrecognizable at 32×32* and used exclusively for educational, non-commercial purposes. Nevertheless, it visually demonstrates how a low-resolution bitmap can be incorporated into the model's parameter space and later *extracted* (possibly with minor distortions) to verify ownership.

## I.2   SPECTRAL DIAGNOSTICS

For each layer we compute the top-$r$ singular values of $\Delta W$ and define $p_i = \sigma_i^2 / \sum_j \sigma_j^2$, the effective rank $r_{\text{eff}} = \exp\left(-\sum_i p_i \log p_i\right)$, and the cumulative energy $S_k = \sum_{i=1}^{k} p_i$. Across backbones, SEAL often shows a lower $r_{\text{eff}}$ and a larger $\Delta S_k := S_k(\text{SEAL}) - S_k(\text{LoRA})$ at small $k$, indicating stronger concentration in early modes as depicted in Figure 8 and 9. This pattern is consistent with the robustness we observe against rank-only obfuscations: when most spectral energy sits in a handful

---

[2]https://www.youtube.com/watch?v=2zHHkSu1br4

of leading directions, truncating tail modes by SVD preserves both task utility and the embedded relation needed by our public test (see Section 5.4 and Figure 3). Empirically, the same bias toward high-energy modes also helps explain why very aggressive parameter removal is required before extraction signals meaningfully degrade under pruning.

## I.3  RANK ABLATION

To evaluate the versatility of the proposed SEAL method under varying configurations, we conducted additional experiments focusing on different rank settings ($r \in \{4, 8, 16\}$). We report the task fidelity for both the deployed passport $C$ (structured bitmap) in Table 19 and the private passport $C_p$ (random Gaussian) in Table 20. The results show that SEAL consistently maintains valid performance levels proportional to the rank capacity, regardless of the passport initialization strategy.

Furthermore, Table 21 presents the watermark robustness against pruning attacks across these ranks. Notably, higher ranks provide a stronger "Defender's Advantage": for $r = 16$, the watermark remains detectable ($-\log_{10} p \gg 3.3$) even when the model is pruned to the point of significant degradation. Conversely, at very low rank ($r = 4$), the capacity for redundancy is lower, yet the watermark survives until the model collapses (13.90% accuracy), rendering the stolen artifact useless.

R:fNRL, P2Pv

Table 19: **Fidelity with Passport $C$ (Bitmap).** Accuracy on commonsense reasoning tasks for SEAL ranks $r \in \{4, 8, 16\}$ using the structured bitmap passport.

| Rank | BoolQ | PIQA | SIQA | HellaSwag | Wino. | ARC-c | ARC-e | OBQA | Avg. |
|---|---|---|---|---|---|---|---|---|---|
| 4 | 63.15 | 76.61 | 71.39 | 66.06 | 67.88 | 60.50 | 79.34 | 68.00 | 69.11 |
| 8 | 64.56 | 79.49 | 74.97 | 75.86 | 73.56 | 65.02 | 82.03 | 73.00 | 73.56 |
| 16 | 65.35 | 80.96 | 77.12 | 81.17 | 75.30 | 65.61 | 81.48 | 74.80 | 75.22 |
| 32 | 66.45 | 82.16 | 78.20 | 83.72 | 79.95 | 68.09 | 82.62 | 79.40 | 77.57 |
| LoRA$_{r=32}$ | 65.96 | 78.62 | 75.23 | 79.20 | 76.64 | 79.13 | 62.80 | 72.40 | 73.75 |

Table 20: **Fidelity with Passport $C_p$ (Random).** Accuracy on commonsense reasoning tasks for SEAL ranks $r \in \{4, 8, 16\}$ using the random Gaussian passport (Base). Performance is consistent with Passport $C$.

| Rank | BoolQ | PIQA | SIQA | HellaSwag | Wino. | ARC-c | ARC-e | OBQA | Avg. |
|---|---|---|---|---|---|---|---|---|---|
| 4 | 63.33 | 76.93 | 71.44 | 65.60 | 68.75 | 60.07 | 79.59 | 68.60 | 69.29 |
| 8 | 64.28 | 79.16 | 74.82 | 75.07 | 74.43 | 65.10 | 81.65 | 75.60 | 73.76 |
| 16 | 65.63 | 80.09 | 76.66 | 81.89 | 76.16 | 66.38 | 82.45 | 76.40 | 75.71 |

Table 21: **Robustness across ranks.** Impact of pruning on task accuracy and watermark detectability. The watermark remains detected ($-\log_{10} p \geq 3.3$) well into the region where the model degrades.

| Rank | Pruning Ratio | Avg. Acc. (%) | Detectability ($-\log_{10} p$) | Outcome |
|---|---|---|---|---|
| | 0.00 (Base) | 69.12 | - | Baseline |
| 4 | 0.80 | 68.72 | 211.39 | Detected |
| | 0.90 | 66.23 | 100.97 | Detected |
| | 0.99 | 13.90 | 0.00 | Removed (Collapsed) |
| | 0.00 (Base) | 73.56 | - | Baseline |
| 8 | 0.80 | 73.80 | 880.33 | Detected |
| | 0.90 | 73.14 | 323.27 | Detected |
| | 0.99 | 22.76 | 0.10 | Removed (Collapsed) |
| | 0.00 (Base) | 75.22 | - | Baseline |
| | 0.80 | 75.73 | 4363.72 | Detected |
| 16 | 0.90 | 75.51 | 1606.09 | Detected |
| | 0.95 | 75.42 | 306.69 | Detected |
| | 0.99 | 63.20 | 72.74 | Detected (Degraded) |

## I.4 Impact of the Size of Passport $C$

To analyze how the magnitude of the passport $C$ influences the final output, we train the model with $\Delta W = B\,C\,A$, but at inference time remove $C$ (i.e., $\mathbb{N}(B, A, \emptyset)$) to observe the resulting images under different standard deviations $\mathtt{std}$ of $C$. Specifically, we sample $C \sim \mathcal{N}(0, \mathtt{std}^2)$ with $\mathtt{std} \in \{0.01, 0.1, 1.0, 10.0, 100.0\}$ and keep $B$ and $A$ trainable. Figure 10 shows that lower $\mathtt{std}$ (e.g., 0.01) produces markedly different images relative to the vanilla model **without** $C$, while higher $\mathtt{std}$ (e.g., 10.0 or 100.0) yields outputs closer to the vanilla Stable Diffusion model[3].

**Why does $\mathtt{std}$ of $C$ affect $\mathbb{N}(B, A, \emptyset)$?** Recall that $\Delta W = B\,C\,A$. If $\mathtt{std}(C)$ is very small (e.g., 0.01), then during training, the product $B\,C\,A$ must still approximate the desired update $\Delta W$. Because $C$ is tiny, $B$ and $A$ tend to have relatively large values to compensate. Consequently, when we *remove* $C$ at inference time (use $\mathbb{N}(B, A, \emptyset)$), these enlarged $B$ and $A$ inject strong perturbations, manifesting visually as high-frequency artifacts.

Conversely, if $\mathtt{std}(C)$ is very large (e.g., 10.0 or 100.0), then to avoid destabilizing training, $B$ and $A$ remain smaller in scale. Hence, removing $C$ at inference, $\mathbb{N}(B, A, \emptyset)$, introduces only minor differences from the original model, leading to outputs that closely resemble the vanilla Stable Diffusion model.

Figure 10: Effect of passport $C$ standard deviation ($\mathtt{std}$) on SEAL weight. $\mathtt{std} = \sigma$: Outputs are using only SEAL weight without $C \sim \mathcal{N}(0, \sigma^2)$, $\mathbb{N}(B, A, \emptyset)$. Vanilla SD 1.5: output from vanilla Stable Diffusion 1.5 with same prompt.

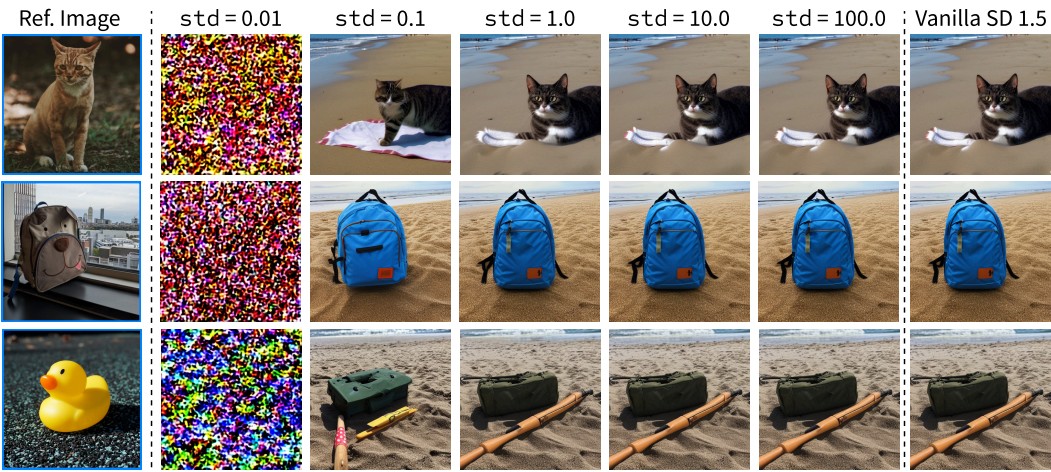

**Quantitative Comparison.** In addition to the qualitative results, Table 22 compares Peak Signal-to-Noise Ratio (PSNR) and Structural Similarity (SSIM) between images generated using only trained SEAL weights **without** $C$, $\mathbb{N}(B, A, \emptyset)$, at various passport $\mathtt{std}$ values. Lower $\mathtt{std}$ (e.g., 0.01) shows significantly lower PSNR and SSIM, indicating large deviations (i.e., stronger perturbations) from the vanilla output. As $\mathtt{std}$ increases to 10.0 or 100.0, the outputs become more aligned with the vanilla model, reflected by higher PSNR/SSIM scores.

---

[3] https://huggingface.co/stable-diffusion-v1-5/stable-diffusion-v1-5. The original weight had been taken down.

Table 22: Comparision of PSNR and SSIM values for images generated **without** $C \sim \mathcal{N}(0, \sigma^2)$, using only $\mathbb{N}(B, A, \emptyset)$, under varying standard deviations of the passport $C$, with images generated under vanilla SD 1.5 model. Obj. 1: `Cat`, Obj 2: `Backpack dog`, Obj. 3: `Ducky toy`. Object names are same as (Ruiz et al., 2023)

| Ref. | Metric ↑ | Standard Deviation of $C$ | | | | |
| | | 0.01 | 0.1 | 1.0 | 10.0 | 100.0 |
|---|---|---|---|---|---|---|
| Obj. 1 | SSIM | 0.104 | 0.691 | 0.936 | 0.987 | 0.998 |
| | PSNR | 7.80 | 19.02 | 30.87 | 43.64 | 53.16 |
| Obj. 2 | SSIM | 0.102 | 0.652 | 0.941 | 0.993 | 0.998 |
| | PSNR | 7.91 | 18.51 | 33.15 | 47.24 | 54.21 |
| Obj. 3 | SSIM | 0.115 | 0.651 | 0.959 | 0.992 | 0.998 |
| | PSNR | 8.08 | 18.39 | 32.92 | 45.39 | 53.58 |

## J  FUTURE WORK

### J.1  MULTIPLE PASSPORTS AND DATASET-BASED MAPPINGS

So far, our main exposition has treated the watermark matrices $C$ and $C_p$, constant passports. However, SEAL naturally extends to a setting in which one maintains multiple passports $\{C_1, C_2, \ldots, C_m\}$ (similarly $\{D_1, D_2, \ldots D_n\}$), each possibly tied to a distinct portion of the training set, or to a distinct sub-task within the same model. Formally, suppose that during mini-batch updates Algorithm 1 randomly picks *one* passport $C_i$ *associated* with $(x, y)$. Then line 4 and 5 of Algorithm 1 becomes:

$$\text{pick } C_i \text{ s.t. } (x, y) \mapsto C_i, \quad W' \leftarrow W + B\,C_i\,A.$$

One can store a simple mapping function $\phi : (x, y) \mapsto i \in \{1, \ldots, m\}$ to tie each batch to its specific passport.

**Distributed / Output-based Scenarios.** Another angle is to use multiple passports not only at *training* time but also during *inference*. For instance, given a family $\{C_1, \ldots, C_m\}$, one could selectively load $C_i$ to induce different behaviors or tasks in an otherwise single LoRA model. In principle, if each $C_i$ is entangled with $(B, A)$, switching passports at inference changes the effective subspace. This may be viewed as a *distributed watermark* approach: where each $C_i$ can be interpreted as a unique "key" that enables (or modifies) certain model capabilities, separate from the main training objective. Though we do not explore this direction in detail here, it points to broader usage possibilities beyond simply verifying ownership, such as controlled multi-task inferences and individually licensed feature sets.

### J.2  BEYOND LOW-RANK ADAPTATION: LINEAR OPERATORS

Although we focused on LoRA-style low-rank updates, the core passport idea (i.e. non-trainable matrices entangled with trainable weights) can apply to general *linear* operators as well. For instance, transformer blocks (query/key) rely on matrix multiplications (Fernandez et al., 2024), where a constant passport could similarly be inserted. Such embedding in broader architectures echoes the functional-layer approach and remains a promising future avenue to combine passports with various advanced parameter-efficient strategies.

