# OpenReview forum: "SEAL: Entangled White-box Watermarks on Low-Rank Adaptation"
_ICLR.cc/2026/Conference — Submitted to ICLR 2026_

### Official Review · Reviewer_WQP7 · 2025-10-29

**Soundness:** 2
**Presentation:** 2
**Contribution:** 2
**Rating:** 2
**Confidence:** 4

**Summary:**

The paper presents a method, SEAL, for training a LoRA adapter with a built in passport watermark. To do so, when training the LoRA module the authors pass the activations through fixed matrices C, C_p between B and A, and optimize them to behave similarly. After training C is decomposed into C=C_1C_2 and the released adapter becomes B'=BC1, A'=C_2A. Given B, A, C, C_p the verification criteria make sure that (i) The released adapter is reconstructed by both BC_pA, BCA and that (ii) the provided verification adapters behave the same.

**Strengths:**

- The task of LoRA watermarking is important and timely.

- The proposed method is simple and does not harm the LoRA training.

- The paper is well-written and easy to follow.

**Weaknesses:**

- **Ambiguity attacks.** My main concern is about further ambiguity attacks of creating new fake passports. The verification criteria only verifies that the claimants have 2 passports that reconstrut the public adapters and behave similarly. Therefore, an adversary could set B=B', A=A', and C=I to obtain a single passport. Then, several possible attacks can be proposed for the second passport:
  - The second passport may be obtainable through optimization on R1 and R2. While the paper argues that adversaries lack the original training data, it remains unclear whether an adversary could optimize C_p on similar auxiliary data to satisfy both reconstruction and fidelity-gap requirements. Given that only C_p requires optimization (with B and A fixed), and that adapter training is very efficient, this represents a plausible attack vector that requires further investigation.
  - As shown in Fig. 2 the effective rank of SEAL adapters is much lower than LoRA adapters. Therefore it should be possible to choose a valid second passport that is orthogonal (or almost orthogonal) to B and A.
  - The verification does not enforce the passports to be different from one another (making several trivial solutions valid).
Therefore, as it stands, I believe that the proposed verification can be fooled with fake passports in a relatively simple way.

- **Missing comparisons.** SEAL is not compared to any prior watermarking approach. It would strengthen the paper to compare SEAL to adaptations of other watermarking approaches which were designed for full fine-tuning.

- **Computation of p is unclear.** The main criteria against the removal, fine-tuning and structural obfuscation attacks is the detectability of the passport. However, I did not find where this criteria is formally defined. This makes it difficult to understand the significance and soundness of the results.


Minor remarks:

- **Qualitative verification.** While the authors present qualitative comparisons of SEAL and LoRA in the appendix, it would also be helpful to visualize the difference when using the 2 passports, i.e., results using BCA, and BC_pA.

**Questions:**

- Given the adapter is optimized with both C and C_p, does SEAL increases the training time of the adapter over LoRA?

---

> ### Author Response · Authors · 2025-11-14
>
> Thank you for the careful and technically detailed review. We will integrate the corresponding changes into a revised manuscript after we can incorporate all reviewers’ feedback coherently.
>
> ---
>
> ### (1) Ambiguity attacks and “fake passports” (including $C=I$)
>
> We fully agree that ambiguity is the core threat, and you correctly point out corner cases like $(B,A)=(B',A')$ with $C=I$, as well as degenerate factorizations. As detailed in our global clarification comment, we will make explicit a strengthened public verification policy by adding a front-end check ($R0$), applied before $R1/R2$:
>
> **R0.a – non-trivial passports.**
> Enforce a minimum separation
> $||C - C_p||_F > \delta_C,$
> where $\delta_C$ is calibrated from the owner’s passport generation rule (e.g., bitmap vs. i.i.d. $\mathcal{N}(0,1)$). Identity-like or nearly identical passports are rejected.
>
> **R0.b – induced weight separation.**
> Enforce a corresponding minimum separation in adapter weight space
> $|\Delta W(C) - \Delta W(C_p)|_F \gt \kappa_0 * \delta_C,$
> where $\kappa_0$ is a lower bound on the smallest singular value of the linear map $C \rightarrow \Delta W(C)$. (See the lemma in global comment.)
>
> **R0.c – shape / rank / conditioning consistency.**
> Require that the submitted $(B, A)$
>
> (i) have the same shapes and ranks as the released LoRA $(B', A')$; and
>
> (ii) satisfy a minimal conditioning constraint $\min_\ell$ {$\sigma_{\min}(B_\ell), \sigma_{\min}(A_\ell)$}$\ge \kappa_0$ across layers.
>
> Under $R0$, the attack pattern you describe is no longer admissible:
> * Claims with $C=C_p$ (or numerically almost identical, including $C=I$) violate $R0.a$/$R0.b$ and are rejected before $R1/R2$.
> * Claims based on an alternative low-rank factorization with aggressive SVD truncation violate $R0.c$ (shape/rank/conditioning mismatch).
>
> We will also fix the phrasing of $R1$ in Sec. 4.2 to match the implementation:
> min{$C_i \in$ {$C,C_p$}} $|\Delta W(C_i) - B'A'|_F \le \rho_T.$
> The current draft incorrectly reads for each $C_i$.
>
> ---
>
> ### (2) Optimization-based attacks on $C_p$ using auxiliary data
>
> We agree that, at first sight, optimizing a second passport $(C_p)$ over auxiliary data to satisfy $R1$ and $R2$ looks like a plausible white-box attack vector.
> Our position is that this attack is computationally unattractive once the full $R0–R2$ policy is enforced:
>
> * **Cost of the objective.** Minimizing the dual-passport gap $\Delta_T$ used in $R2$ requires **two full evaluation passes per step** (for $C$ and $C_p$), unlike standard LoRA training which updates a single adapter via a training loss. Matching our reported gaps on large evaluation suites becomes significantly more expensive than just training a new adapter.
> * **Tension with $R0$.** To pass $R2$, an attacker wants $C_p$ to be functionally close to $C$; to pass $R0.a$/$R0.b$, they must keep both matrix-level and induced-weight-level separation above $\delta_C$ and $\kappa_0*\delta_C$. This sharply shrinks the feasible region for any optimization over $C_p$.
> * **No access to the original co-training dynamics.** Our dual passports are entangled with $(B,A)$ during training. Post-hoc optimization over $C_p$ alone, with fixed $(B',A')$, cannot reproduce the same joint trajectory we used to obtain entanglement, so the attacker must essentially solve a harder inverse problem under tight constraints rather than “just fine-tune a LoRA”.
>
> We will clarify this attack model and the above trade-offs in the security discussion.
>
> ---

---

> > ### Author Response · Authors · 2025-11-14
> > **continued comments**
> >
> > ### (3) Effective rank and “orthogonal” fake passports
> >
> > Thank you for raising the connection with the effective rank plots. It is important to clarify what they do not imply.
> > The relation
> > $\Delta W(C) = B C A$
> > is the outcome of co-training $B$ and $A$ with a fixed passport $C$; it is not an arbitrary post-hoc factorization. An attacker cannot freely choose a new passport $(C_p)$ “orthogonal to $B$ and $A$” in some basis and expect it to pass the verification rule.
> >
> > This is where the conditioning lemma underlying $R0.b$ becomes relevant. Vectorizing,
> > $vec(\Delta W(C)) = (A^T \otimes B) * vec(C),$
> > whose singular values are all products $\sigma_i(A) \sigma_j(B)$. If we define
> > $\kappa_0 = min_l (\sigma_\min(B_l) * \sigma_\min(A_l)) \gt 0,$
> > then for any passports $(C, C_p)$,
> > $|\\Delta W(C) - \\Delta W(C_p)|_F \\ge \\kappa_0 * |C - C_p|_F.$
> >
> > Empirically, for our Gemma-2B SEAL adapter (commonsense benchmark), we measure across 90 LoRA layers
> > $(\kappa_\min \approx 4.2 \times 10^{-4})$, which gives a concrete, strictly positive lower bound. Thus:
> >
> > * A $C_p$ designed to be “orthogonal” or very different from $C$ while still satisfying $R0.a$ inevitably induces a non-negligible weight difference via the inequality above, which almost certainly breaks $R2$ because $(B,A)$ were never trained with that passport.
> > * A $C_p$ engineered to keep the model behavior close enough to pass $R2$ must, by the same inequality, be close to $C$ in matrix space, at which point it is rejected by $R0.a$/$R0.b$.
> > * Any attempt to bypass this by changing the factorization $(B,A)$ (e.g., via SVD truncation or re-ranking) will violate $R0.c$ (shape/rank/conditioning mismatch). We will move this lemma closer to the policy definition and explicitly connect it to the reviewer’s “orthogonal passport” concern.
> >
> > ---
> >
> > ### (4) Missing comparisons and definition of the detection `p`-value
> >
> > **Comparisons.** We agree that the connection to prior watermarking work should be clearer. As we explain in more detail in the paper and global comment:
> >
> > * Existing passport methods are designed around **normalization layers** (e.g., BatchNorm / GroupNorm). Standard LoRA adapters, by construction, **do not include these layers** along the adapter path; inserting them “just for comparison” would change the very object we claim to watermark.
> > * Our focus is therefore **“watermarking for LoRA”** (protecting the adapter as IP), rather than using LoRA as a tool to watermark a full model.
> > * Because of this architectural mismatch, we provide a **qualitative comparison** with representative passport and white-box watermarking methods in the related-work/appendix, and focus the quantitative evaluation on a broad suite of white-box attacks (removal, continued fine-tuning, structural obfuscation, ambiguity). We will make this positioning more explicit in the main text so the lack of direct numerical baselines does not come across as an omission.
> >
> > **$p$-value.** You are absolutely right that the definition of the detection criterion needs to be fully explicit.
> > In the revised manuscript we will:
> >
> > 1.  Introduce the detection statistic **once, in the verification section** (before the attack experiments), and
> > 2.  Clearly state that we compute a **two-sided Binomial test** over sign agreements between the original and recovered passport entries, under a null of match probability 0.5, and report $-log_{10}(p)$ for numerical stability.
> >
> > This is already the statistic we use in the pruning and obfuscation experiments; we will simply move the formal definition and notation earlier so that the meaning of “detectable” is unambiguous.
> >
> > ---
> >
> > ### (5) Training-time overhead and qualitative visualization
> >
> > Regarding your question on training time: yes, SEAL does increase adapter training time relative to plain LoRA, but is comparable to other enhanced adapters.
> > On our LLaMA-2 commonsense benchmark (4x RTX 3090, batch size 4), we measure:
> >
> > **Wall-clock training time**
> > * LoRA: 12.0 h, `(81.67 +/- 1.03)`
> > * DoRA: 18.5 h, `(81.98 +/- 0.26)`
> > * SEAL: 19.6 h, `(83.78 +/- 0.27)`
> >
> > So SEAL is roughly +63% over LoRA, and only about +6% over DoRA, while improving accuracy by +2.1 pp over LoRA and slightly over DoRA. At inference, SEAL has no extra cost: the released adapter is just $(B',A')$.
> >
> > We will move these numbers (and a short discussion of memory overhead, which is ~2% in our largest setting) into the main text to give a concrete sense of the overhead.
> >
> > Finally, we appreciate your suggestion on qualitative verification. In the appendix of the revised version we will add visual comparisons of outputs produced via the two passports (i.e., with $BCA$ vs. $BC_pA$) on representative tasks, to illustrate that they are indeed functionally similar despite being distinct in parameter space.
> >
> > Thank you again for your detailed critique. Your comments directly helped us tighten both the verification policy and the way we present SEAL’s security guarantees.

---

> ### Author Response · Authors · 2025-11-23
> **Revisions Uploaded**
>
> Thank you again for your rigorous security analysis. We have uploaded a revised manuscript that addresses your concerns regarding ambiguity attacks and evaluation metrics.
>
> Summary of Changes for You:
> 1. **Ambiguity Defense**: Expanded Appendix E.4 and Sec. 4.2 to explain why optimizing a fake passport is computationally prohibitive due to the tension between R0 (structural separation) and R2 (functional equivalence).
> 2. **Qualitative Verification**: Added Appendix H.4 (Figure 6), visually confirming that the distinct passports ($C$ and $C_p$) generate indistinguishable results.
> 3. **Statistical Rigor**: Explicitly defined the detection statistic ($-\log_{10}p \ge 3.3$) in Sec. 5.4.
> 4. **Comparison Logic**: Clarified in Sec. 2.4 why direct comparison with normalization-based watermarks is architecturally invalid for LoRA.
>
> To facilitate your check, we have highlighted these specific revisions in blue and marked them with the tag **WQP7** in the right margin of the updated PDF. We hope this clarifies the robustness of our method.

---

### Official Review · Reviewer_P2Pv · 2025-10-31

**Soundness:** 2
**Presentation:** 2
**Contribution:** 2
**Rating:** 2
**Confidence:** 3

**Summary:**

This paper introduces SEAL, a novel white-box watermarking scheme designed specifically to protect the intellectual property (IP) of Low-Rank Adaptation (LoRA) adapters. The core problem it addresses is verifying ownership of the adapter weights themselves, which are increasingly trained and shared as standalone IP. SEAL's mechanism is based on "entangled dual passports". During fine-tuning, two fixed, non-trainable passport matrices are inserted between the LoRA matrices. The main contribution is a public verification protocol. To claim ownership, a claimant submits their original, pre-factorized weights and their two passports. The authors provide empirical evidence across LLMs, VLMs, and text-to-image models, showing that SEAL maintains task fidelity while resisting common attacks like pruning, continued fine-tuning, structural obfuscation, and ambiguity attacks.

**Strengths:**

- The paper's primary strength is its significant problem formulation. While DNN watermarking is well-studied, most methods target the entire base model or rely on black-box outputs. This work correctly identifies that for the PEFT ecosystem, the adapter itself is the distributable IP. Defining a white-box, adapter-level ownership verification protocol  is a practical contribution.
- The paper is exceptionally well-written. The method, threat model, and verification protocol are all defined formally and clearly.

**Weaknesses:**

- The core defense against ambiguity attacks is the dual-passport fidelity gap, for which the paper proposes a formal statistical guarantee using Hoeffding's inequality. However, in the paper's own experiments, this formal guarantee fails for the Mistral-7B model, where the owner's observed gap far exceeds the theoretical threshold.
- The limitations section admits that "An adversary who re-trains on similar data may reproduce the owner's dual entanglement and pass verification by design". The threat model assumes the adversary lacks the fine-tuning data. However, a huge number of popular, shared LoRA adapters are fine-tuned on public datasets. In this very common scenario, an adversary has the model, the (public) SEAL method, and the (public) data. They could simply re-train their own adapter with their own passports, achieving similar performance and creating a valid-looking claim. This severely limits the practical utility of SEAL to only those adapters trained on proprietary datasets.

**Questions:**

- The failure to achieve the formal statistical guarantee for Mistral-7B (Table 1) is a key concern. Why does the dual-passport gap $\hat{\Delta}_T$ become so large for this specific model? Does this failure correlate with the LoRA rank $r$, model architecture, or task? How sensitive is the gap to these hyperparameters?
- The method seems insecure when the fine-tuning data is public, as an adversary can just replicate the training process with their own passports. Am I misunderstanding the threat model, or does SEAL only offer protection for adapters trained on proprietary datasets? If so, this seems like a major limitation that should be stated more clearly.
- To confirm my understanding of the verification protocol: Does the owner need to privately store the pre-factorized weights $(B, A)$ in addition to the private passport $C_p$ to make a future claim? If so, this implies a 2x storage overhead for the owner's proof, correct?
- How sensitive is the method's performance (both task fidelity and robustness) to the choice of $C$ and $C_p$? What happens if $C$ and $C_p$ are chosen to be (a) (pseudo-)orthogonal, (b) very similar (small $||C - C_p||_F$), or (c) low-rank?

---

> ### Author Response · Authors · 2025-11-14
>
> Thank you for your detailed and candid review.
> For points that require additional experiments or substantial edits to the main text, we will first respond here
> and then integrate the changes in a revised manuscript after we can incorporate all reviewers’ feedback coherently.
>
> ---
>
> ### (1) Formal FPR guarantee and the Mistral‑7B
>
> You are right that for Mistral‑7B on the large commonsense suite (Table 1), the observed dual‑passport gap
> exceeds the Hoeffding‑based threshold, so the formal false‑positive guarantee does not apply there.
>
> As we will clarify in the revision:
>
> - The Hoeffding bound assumes i.i.d. Bernoulli trials with homogeneous variance. In this particular Mistral‑7B setting,
>   $M_T$ aggregates a heterogeneous suite (different benchmarks and difficulties), so these assumptions are only approximate.
> - In most other settings the empirical gap stays well below the bound; Mistral‑7B is a “stress‑test” where the
>   bound becomes slightly optimistic.
>
> We hypothesize this deviation is also linked to model-specific instability. Unlike the other models, Mistral-7B required LR tuning even for standard LoRA to converge properly. While SEAL did train successfully, this underlying instability likely contributed to the larger observed gap.
>
> ---
>
> ### (2) Open‑data scenario and practical utility
>
> This is a critical point about our threat model , and we appreciate the chance to clarify our IP definition.
>
> You are correct: If an adversary uses **public data** and their **own compute** to train a new SEAL adapter with their own passports, that is **their model**. Our method does not—and should not—claim ownership over it. What SEAL does protect is the specific trained artifact (e.g., `.pt` file)—the result of the owner's compute cost and unique training run (e.g., their 'own recipe'). Your concern ("insecure when data is public") implies an attacker can replicate our unique entanglement just by re-training. This is empirically false. Our Table 5 (Finetuning Attack) already simulated this: retraining on the same public data (commonsense reasoning) ($C_{3e} \\rightarrow C_{1e}$) failed to erase or overwrite the original watermark. An attacker cannot fabricate our specific $(B,A,C,C_p)$ bundle post-hoc. As discussed in our Global Comment (Sec 2), this is mathematically contradictory (violates **R0**) and economically non-viable (due to the computational cost of forging **R2**).
>
> We will clarify this limitation: SEAL protects **the compute and the resulting artifact**, not the idea of training on public data.
>
> ---
>
> ### (3) Storage overhead: “2× weights” for the owner
>
> Your understanding is essentially correct:
>
> - The **public** artifact is just the folded adapter $(B',A')$ (LoRA‑like size).
> - The **owner** must keep a proof bundle consisting of the pre‑factorized $(B,A)$ and the two passports $(C,C_p)$.
>
> This is roughly a 2× storage overhead on the adapter, but still **tiny relative to the base model**. In our view this is a realistic trade‑off: This is analogous to keeping a backup or a signed original of a contract in a safe. Given that LoRA adapters are already small, storing this 'proof' bundle is a negligible and standard cost for verifiable IP protection. LoRA adapters are already small, and storing a second copy plus two $r\\times r$ matrices is a negligible cost for verifiable ownership.
>
> ---
>
> ### (4) Sensitivity to passport geometry (orthogonal / similar / low‑rank)
>
> Thank you for pushing on this point. Our strengthened policy and lemma speak directly to these cases:
>
> - If $C$ and $C_p$ are **too similar** ($\\|C-C_p\\|_F \\le \\delta_C$), they are rejected by R0.a.
> - If they are numerically similar in *weight space* ($\\|\\Delta W(C)-\\Delta W(C_p)\\|_F \\le \\kappa_0 \\delta_C$),
>   they are rejected by R0.b.
> - If they are “degenerate” (e.g., low‑rank) in a way that relies on collapsing singular values, this typically
>   violates the minimal conditioning requirement in R0.c.
>
> Our lemma shows that for any two passports,$|\Delta W(C)-\Delta W(C_p)|_F$
> $\\ge \\kappa_0 |C-C_p|$ and $\kappa_0 \gt 0$ determined by the per‑layer singular values of $(B,A)$. Empirically, for our Gemma‑2B adapter we measure
>
>  $\kappa_{\min}\approx 4.2\times 10^{-4}$
>  across 90 layers, giving a concrete lower bound.
> Thus, orthogonal or low‑rank passports that still satisfy R0 necessarily induce a non‑negligible separation in weight space,
> and this separation is what R2 exploits when we test the dual‑passport gap. In the revision we will:
>
> - move this lemma (with intuition) closer to Sec. 4, and
> - make the constraints on $(C,C_p)$ explicit in the public rule, so that the role of “non‑trivial, well‑conditioned passports”
>   is clear. Empirically, we also note in Tab. 2 that the i.i.d. $\\mathcal{N}(0,1)$ passport—a "DL-friendly" distribution—achieved the highest task fidelity.
>
> Thank you again for raising these conceptual issues—they helped us sharpen both the policy and the positioning of SEAL’s guarantees.

---

> ### Author Response · Authors · 2025-11-23
> **Revisions Uploaded**
>
> Thank you again for your critical and insightful review. We have uploaded a revised manuscript that sharpens the threat model and verification logic based on your comments.
>
> Summary of Changes for You:
> 1. **Open Data & Threat Model**: Explicitly distinguished between "recipe replication" (retraining) and "artifact forgery" in Sec. 2.3 and Limitations, clarifying that SEAL protects the specific trained weights.
> 2. **Theoretical Analysis**: Added Appendix E (including Lemma 1 proof and empirical validation of $\kappa_0$) to provide rigorous backing for the R0 policy.
> 3. **Mistral-7B**: Clarified the "empirical-only" guarantee for non-i.i.d. data in Sec. 4.3.
>
> To facilitate your check, we have highlighted these specific revisions in blue and marked them with the tag **P2Pv** in the right margin of the updated PDF. We believe these edits solidify the paper's claims.

---

### Official Review · Reviewer_fNRL · 2025-10-31

**Soundness:** 3
**Presentation:** 2
**Contribution:** 2
**Rating:** 4
**Confidence:** 2

**Summary:**

This work proposes SEAL, a universal white-box watermarking scheme for LoRA weights. To address the ownership protection issue of LoRA weights, the scheme inserts non-trainable main passports and auxiliary passports between the trainable matrices of LoRA. Experiments demonstrate that SEAL causes no performance degradation across various tasks and even outperforms standard LoRA in some scenarios.

**Strengths:**

1.	This paper designs a white-box watermarking mechanism specifically for the LoRA structure to facilitate the ownership protection of relevant weights.
2.	The method is concise and can maintain model performance across multiple experimental scenarios.

**Weaknesses:**

1. This work does not explore the impact of matrix properties such as distribution and sparsity on watermark robustness and model performance, nor does it systematically compare the effectiveness differences between different types of passports. The design basis remains insufficient.

2. The time overhead introduced by this method is still significant (Table 8), and it does not quantify the memory consumption of SEAL during training or the additional overhead during inference compared to standard LoRA.

3. Verification has only been conducted under traditional attack scenarios such as pruning. Can experiments be performed under more advanced attack scenarios currently available?

4. When the rank of LoRA changes, the dimension of the passport matrix needs to be adjusted synchronously. Will this adjustment affect the embedding effect and extraction accuracy of the watermark? Can curves showing the changes in SEAL watermark robustness and model performance under different rank settings be provided?

**Questions:**

With reference to the Weaknesses section.

---

> ### Author Response · Authors · 2025-11-14
>
> Thank you for your careful review and constructive questions.
> For points that require additional experiments or substantial edits to the main text, we will first respond here
> and then integrate the changes in a revised manuscript after we can incorporate all reviewers’ feedback coherently.
>
> ---
>
> ### (1) Design space of passports (distribution, sparsity, types)
>
> Thank you for raising this point about the design basis. Our current experiments focus on **dense, full‑rank passports**, and we agree that a more systematic exploration is an important direction.
>
> - **What we already varied.**
>   In Sec. 5.2 and Appendix I.1 we compare two concrete families: a structured bitmap passport and an i.i.d. Gaussian passport sampled from $\\mathcal{N}(0,1)$. Across LLM / VLM / T2I settings, these two choices yield *very similar* task fidelity and robustness, which suggests that SEAL is not overly sensitive to the choice of dense, full‑rank distribution.
>
> - **Why we did not yet sweep sparsity / low rank.**
>   Exploring structured sparsity or low‑rank passports in a principled way would require re‑running our full cross‑modal benchmark with multiple passport families, which is beyond what we could fit into this submission. Instead, we chose to first show that one simple scheme (dense, full‑rank passports) is robust and practical across modalities.
>
> - **Design constraints going forward.**
>   In the global clarification comment we introduce an explicit policy‑level check $(R0)$ (non‑trivial separation and minimal conditioning), which any passport family must satisfy. In the revised manuscript we will state this more clearly and position a broader exploration of distribution / sparsity / rank as an explicit piece of future work built on top of the $(R0)$ constraints.
>
> ---
>
> ### (2) Training time and memory overhead
>
> Thank you for prompting us to quantify this more clearly.
>
> On the LLaMA‑2 commonsense setting (4× RTX 3090, batch size 4), we measure:
>
> - **Peak training memory (per GPU)**
>   - LoRA: $\\approx 16.80$ GB
>   - SEAL: $\\approx 17.13$ GB
>
>   That is, SEAL adds roughly **0.3 GB (~2%)** over standard LoRA in training memory, with no change in parameter count.
>
> - **Wall‑clock training time (same setting, from Appendix, Table 9)**
>
>   - LoRA: 12.0 h, 81.67$\pm$1.03
>   - DoRA: 18.5 h, 81.98$\pm$0.26
>   - **SEAL**: 19.6 h, 83.78$\pm$0.27
>
> So relative to LoRA, SEAL incurs roughly +63% training time, comparable to (and slightly above) DoRA, while delivering +2.1 pp better accuracy than LoRA and slightly better than DoRA. We will move these numbers into the main text and add a short discussion that SEAL trades a moderate training‑time overhead for (i) verifiable ownership and (ii) a small accuracy gain.
>
> At inference time, runtime and memory are essentially identical to LoRA, since the public adapter is just \\((B',A')\\) with the passport already folded in.
>
> ---
>
> ### (3) “Advanced” attack scenarios and threat model scope
>
> We apologize for the wording that made it sound as if we only tested pruning. In fact, our *verification rule* (now stated as **R0-R2 **in the global comment) is evaluated under four white‑box attack classes in Sec. 5.4:
>
> - removal / sparsification (adapter pruning, SVD factorization),
> - continued fine‑tuning on downstream data,
> - structural obfuscation (rank change / refactorization), and
> - ambiguity attacks via interpolated or optimized counterfeit passports.
>
> These are, to our knowledge, the most relevant **white‑box** attacks when the adversary already has access to the adapter weights.
>
> We suspect that by “more advanced attack scenarios” you may also have in mind powerful **black‑box** attacks such as distillation or model‑generated data. In our setting, however, the adversary already has the weights and can fine‑tune directly (which we do test), so such black‑box attacks are less natural than in full‑model watermarking. In the revision we will:
>
> - make this white‑box assumption explicit in the threat‑model section, and
> - more clearly organize Sec. 5.4 around these four attack families so the scope of our evaluation is easier to see.
>
> ---
>
> ### (4) Rank changes and robustness vs. LoRA rank
>
> We agree this is an important practical question. When the LoRA rank $r$ changes, the passport dimension must change with it, and robustness could in principle depend on $r$.
>
> In a revised version we will therefore:
>
> - run an additional experiment on a representative model (e.g., Gemma‑2B) with multiple LoRA ranks $r \\in \\{4,8,16,32\\}$, and
> - report curves showing how both **task performance** and **watermark detectability** (e.g., \\(-\\log_{10}(p)\\) under pruning) vary with \\(r\\).
>
> We will also comment explicitly on whether we observe any trade‑off between higher rank (more capacity) and watermark robustness in this setting.
>
> Thank you again for these constructive suggestions—they directly helped us clarify the design space, the cost profile, and the evaluation scope of SEAL.

---

> ### Author Response · Authors · 2025-11-17
> **Response to Rank Ablation & Watermark Robustness (Table A & B)**
>
> To address your question about how changing the LoRA rank $r$ (and thus the passport dimension) affects watermark embedding and extraction, we ran an additional ablation on Gemma‑2B with ranks $r \in \{4, 8, 16\}$. In short, we observe that SEAL continues to embed and extract the watermark reliably across all tested ranks, with a predictable robustness–capacity trade‑off.
>
> ## 1. Establishing Verification Criteria (Table A)
> First, we establish the baseline **Fidelity Gap ($\Delta_T$)** for the Dual-Passport Protocol (R2) across different ranks. As shown in **Table A**, the gap remains consistently small ($< 0.5\%$) across ranks, so the “entanglement” property is preserved even when we resize the passports.
>
> _Table A. Reference Fidelity Gap ($\Delta_T$) across LoRA Ranks. ($M_T$: Accuracy on Commonsense QA)_
> | Rank ($r$) | $C_a$ (bitmap) Acc. | $C_b$ (normal) Acc. | Reference Gap ($\Delta_T$) |
> |:---:|:---:|:---:|:---:|
> | 4 | 69.12 | 69.29 | **0.17** |
> | 8 | 73.56 | 73.76 | **0.20** |
> | 16 | 75.22 | 75.71 | **0.49** |
>
> ## 2. Detection Metric (p-value)
> * **Statistic:** $- \log_{10} p$ (Two-sided hypothesis test).
> * **Threshold:** $\ge 3.3$ corresponds to $p \le 5 \times 10^{-4}$ (Significant Detection).
>
> ## 3. Robustness and Defender's Advantage (Table B)
> **Table B** illustrates the trade-off an attacker faces under pruning attacks.
> * **Defender's Advantage:** In significant pruning scenarios (e.g., Rank 16 @ 99%), the watermark remains **Detected** even though the model performance is **Degenerated** (dropped by >10%). This implies the attacker failed to secure a usable model without triggering the watermark.
> * **Total Collapse:** When the watermark is finally removed (e.g., Rank 4 @ 99%), the model performance has **Collapsed** (Acc $\approx$ 13%), rendering the stolen artifact useless.
>
> _Table B. Impact of LoRA Rank ($r$) on Watermark Robustness against Pruning Attacks._
> | Rank ($r$) | Pruning Ratio | Avg. Acc. (%) | Detectability ($- \log_{10} p$) | Outcome |
> |:---:|:---:|:---:|:---:|:---|
> | **4** | 0.0 (Base) | 69.12 | - | Baseline |
> | | 0.80 | 68.72 | 211.39 | Detected |
> | | 0.90 | 66.23 | 100.97 | Detected (Degenerated) |
> | | 0.95 | 56.12 | 48.89 | Detected (Degenerated) |
> | | 0.99 | 13.90 | 0.00 | Watermark removed, model collapsed |
> | **8** | 0.0 (Base) | 73.56 | - | Baseline |
> | | 0.80 | 73.80 | 880.33 | Detected |
> | | 0.90 | 73.14 | 323.27 | Detected (Degenerated) |
> | | 0.95 | 70.59 | 117.57 | Detected (Degenerated) |
> | | 0.99 | 22.76 | 0.10 | Watermark removed, model collapsed |
> | **16** | 0.0 (Base) | 75.22 | - | Baseline |
> | | 0.80 | 75.73 | 4363.72 | Detected |
> | | 0.90 | 75.51 | 1606.09 | Detected |
> | | 0.95 | 75.42 | 306.69 | Detected |
> | | **0.99** | **63.20** | **72.74** | **Detected (Degenerated)** |
> | | 0.999 | 7.16 | 0.00 | Watermark removed, model collapsed |
>
> Overall, these results indicate that:
> - resizing the passport with rank does not harm watermark embedding or extraction, and
> - higher ranks naturally provide more headroom: the attacker must either accept strong watermark detectability or prune so aggressively that the model becomes unusable.
>
> We will add these ablation tables (with full details and plots) to the appendix in the revised manuscript.

---

> ### Author Response · Authors · 2025-11-23
> **Revisions Uploaded**
>
> Thank you again for your practical and detailed feedback. We have uploaded a revised manuscript containing the additional experimental data you requested.
>
> Summary of Changes for You:
> 1.  **Training Overhead**: Added Appendix D.3 (Table 8), quantifying the training time (+63%) and memory (+2%) costs, which are comparable to other advanced adapters.
> 2.  **Rank Ablation**: Added Appendix I.3 (Tables 19-21), demonstrating that watermark robustness and task fidelity remain consistent across ranks $r \in \{4, 8, 16\}$.
>
> To facilitate your check, we have highlighted these specific revisions in blue and marked them with the tag **fNRL** in the right margin of the updated PDF. We hope these results satisfy your inquiry on efficiency and scalability.

---

### Official Review · Reviewer_d1zQ · 2025-11-01

**Soundness:** 3
**Presentation:** 2
**Contribution:** 3
**Rating:** 6
**Confidence:** 3

**Summary:**

SEAL proposes a white-box watermark for LoRA adapters. During training, two fixed matrices (“passports”) \(C\) and \(C_p\) are inserted between \(B\) and \(A\), and the model alternates them per mini-batch so both versions behave almost identically (they’re “entangled”). After training, \(C\) is factorized and folded into \((B',A')\) so the released adapter looks like a normal LoRA (no extra runtime cost).
*Public verification* accepts a claim if and only if:
- **R1 (reconstruction):** the submitted passports reproduce the released adapter within a small tolerance,
- **R2 (behavioral gap):** the performance difference between the two passports is below a predeclared threshold \(\tau_T\) (for accuracy metrics, \(\tau_T\) can be set via Hoeffding with a target false-positive rate \(\alpha_T\)).

They evaluate the method on several language, vision-language, and diffusion tasks, and test how well the watermark holds under typical model modifications such as pruning, additional fine-tuning, or re-factorization.

**Strengths:**

1. **Clear and simple idea.**
The proposed mechanism is straightforward: alternate two fixed matrices during training and fold one of them into the released adapter. The approach is easy to implement, requires no architectural changes, and adds no inference overhead.

2. **Clearly specified verification rule (but missing one policy detail).**
The paper precisely describes the public verification procedure through two checks (R1 and R2), and for accuracy-based tasks derives the threshold \(\tau_T\) from Hoeffding’s inequality, giving a clear statistical false-positive rate. However, the current rule still lacks an explicit safeguard against trivial claims (e.g., using identity passports), which should be clarified to make the verification policy complete.

3. **Empirical validation of the verification process.**
The authors do not just describe R2 but actually test it quantitatively. Across multiple tasks, the measured gaps are small, and cases where the bound does not hold are explicitly marked as “empirical-only” rather than ignored.


4. **Broad evaluation and robustness analysis.**
The method is tested on several language, vision-language, and diffusion adapters, and under model modifications such as pruning, continued fine-tuning, and re-factorization. The watermark remains detectable under all reasonable perturbations.

5. **Clear presentation and transparency.**
The paper is generally easy to follow, the decision rule and thresholds are explicitly stated, and an anonymized repository is announced for reproducibility. The authors are transparent about the limits of their guarantees and mark where assumptions may break (e.g., “empirical-only” cases). However, some parts could be improved in terms of clarity and positioning—for instance, the *Spectral Diagnostics* section interrupts the flow—and a short explanation of *why* the theoretical bound fails for the Mistral-7B experiment would make the presentation more complete.

6. **Addresses a real and underexplored need.**
While white-box watermarking is less practical than black-box verification, focusing on LoRA adapters fills an important gap and could inform more general ownership verification methods in the future.

**Weaknesses:**

While the paper is well-written and the results are convincing, several aspects could be clarified or strengthened:

1. **Public rule can accept trivial claims unless policy forbids them.**
   Someone could submit \((B,A)=(B',A')\) and set both passports to the identity matrix. That would pass **R1** (exact reconstruction) and **R2** (zero gap) automatically. If I am not mistaken, the paper does not explicitly say the verifier should reject such submissions or require prior provenance/commitment from the owner. This should be stated clearly in the verification policy.
2. **“Empirical-only” acceptance appears in at least one important case.**
   For Mistral-7B on the large commonsense suite, the measured gap is above the Hoeffding cutoff, so the formal false-positive guarantee does not apply (acceptance is operational). A short guideline on choosing \(N_T\) and \(\alpha_T\) to keep acceptance formal would help.
3. **Open-data setting weakens the method against determined attackers.**
   If adapters are trained on fully open-source datasets, an attacker can access the same or very similar data (or sample from the model in the generative case) and *retrain* their own SEAL adapter with correlated passports. With enough compute or queries, this might allow reproducing the behavioral entanglement and pass R2. Even access to a *small* amount of in-distribution data may enable *optimization-based attacks* that search for passports (or for \((B,A,C,C_p)\) tuples) that satisfy R1 and reduce the R2 gap - so the attacker need not always match the defender’s full compute budget. This limitation is worth emphasizing because many LoRA releases rely on public data. However, one has to admit that watermarking always relies on a practical asymmetry: the attacker typically does not want to invest as much compute as the defender—otherwise they could simply train their own model from scratch.

4. **Placement/clarity of spectral diagnostics.**
   The *Spectral Diagnostics* subsection is interesting but at the same time confusing, and thus interrupts the flow at the start of experiments. Consider moving/expanding details to/in the appendix or later in the section, and add 1–2 sentences explaining how these diagnostics feed into the main results.

5. **Relevance of white-box watermarking.**
   White-box watermarking assumes access to the adapter weights and verification via internal parameters, which limits its practical relevance compared to black-box methods. Still, it is a valid and growing research branch that helps understand and strengthen ownership verification under full model access.

**Questions:**

1. **Attacks using generated or similar data.**
   The paper mentions that retraining on similar data could, in principle, reproduce the watermark “by design.” Could you expand on how serious this risk is in practice?
   For example, could an attacker use model-generated samples, a well-trained generative model, or a distillation-style attack to approximate the training distribution and either (a) optimize for new passports that pass R1 and R2 or (b) retrain a SEAL adapter that reproduces the entanglement?
   A short qualitative discussion of how realistic such data-driven or distillation-based attacks are would help clarify the limits of SEAL’s robustness.

2. **Decision rule corner case / provenance.**
   As written, Sec. 4.2 (R1 + R2) would accept a trivial claim like \((B,A)=(B',A')\) with \(C_a=C_b=I\). Am I missing something?
   If not, will you explicitly forbid identical or trivial passports in the public rule to validate ownership?  Please clarify what a verifier *must* demand in practice.

3. **Why the formal FPR bound fails for Mistral-7B.**
   In Sec. 4.3, the false-positive rate is controlled using Hoeffding’s inequality.
   For Mistral-7B, the measured gap is larger than the theoretical bound. Could you explain why this happens and which assumption might not be fulfilled in this case?

---

> ### Author Response · Authors · 2025-11-14
>
> Thank you very much for the thoughtful and technically sharp review. For points that require additional experiments or substantial edits to the main text, we will first respond here and then integrate the changes in a revised manuscript after we can incorporate all reviewers’ feedback coherently.
>
> ---
>
> ### (1) Public rule, trivial claims, and provenance
>
> We fully agree with your concern that, as *written*, the public rule could accept trivial claims such as
> $(B,A)=(B',A')$ with $C=C_p=I$.
> As detailed in our global comment, we now **make explicit** a policy‑level safeguard $R0$:
>
> - $R0.a$: enforce a non‑trivial separation $\|C-C_p\|_F > \delta_C$, calibrated from the owner’s passport generation rule
>   (e.g., bitmap vs. i.i.d. $\mathcal{N}(0,1)$).
> - $R0.b$: enforce a corresponding minimum separation in weight space
>   $\|\Delta W(C)-\Delta W(C_p)\|_F > \kappa_0 \delta_C$ using the singular‑value lemma.
> - $R0.c$: require $(B,A)$ to match the **shape, rank and conditioning** of $(B',A')$.
>
> Under $R0$, a claim with $C=C_p$ (including identity) is rejected *before* R1/R2, and any degenerate refactorization
> obtained by SVD truncation is rejected by a shape/rank/conditioning mismatch.
>
> We will revise Sec. 4.2 to:
>
> - state $R0$ explicitly as part of the **public verification policy**, and
> - clarify that a verifier must *by policy* reject identity‑like or numerically indistinguishable passports and
> rank‑destroying factorizations.
>
> ### (2) “Empirical‑only” acceptance and the Mistral‑7B case
>
> You are right that the observed 3.70 pp gap for Mistral-7B exceeds the 1.09 pp theoretical cutoff . This is not a failure of the protocol, but it highlights why the "empirical-only" rule  is a necessary safety net.
> Two points we will clarify:
>
>   - **Why Hoeffding Fails**: The Hoeffding inequality 4 assumes a strict i.i.d. setting. Our commonsense benchmark ($M_T$) violates this, as it aggregates several heterogeneous benchmarks (BoolQ, PIQA, etc.).
>
>   - **Why "Empirical" is Necessary**: For such real-world, non-i.i.d. datasets, the theoretical bound can be an optimistic approximation. The "empirical-only" rule is the pragmatic fallback that accepts the true observed gap ($\hat{\Delta}_T$), correctly preventing the protocol from falsely rejecting the legitimate owner.
>
> In addition, we will add the requested **practical guideline** in Sec. 4.3: To claim a formal $FPR \le \alpha_T$, the observed owner gap $\hat{\Delta}_T$ (on a validation split of size $N_T$) **must** be below the Hoeffding bound $\tau_T^{theory}$. If this check fails—as in the Mistral-7B stress-test—the protocol explicitly requires falling back to the empirical-only rule ($\tau_T = \hat{\Delta}_T + \eta_T$), and no formal guarantee is claimed.
>
> ### (3) Open‑data setting and attacks using similar or generated data
>
> We agree that the open‑data regime is an important and common setting, and that it weakens any watermark
> that relies on data asymmetry.
>
> Our position (and what we will emphasize more clearly):
>
> - If the fine‑tuning data are fully public and an adversary is willing to invest **comparable compute**, they can
>   indeed re‑train *their own* SEAL adapter with their own passports on the same data and obtain a valid competing claim.
>   This is inherent: given the same base model, method, and data, we cannot prevent multiple independent owners.
>
> - What SEAL *does* guarantee is that a claimant must present a pair $(C,C_p)$ and pre‑factorized $(B,A)$ that
>   were actually co‑trained; under $R0$ + $R1$ + $R2$, it is computationally unattractive to *fabricate* such a pair
>   ex post by optimization on a small auxiliary set.
>
> - Distillation‑style or model‑generated‑data attacks fall into the same bucket: they are, in effect, retraining a new
>   SEAL adapter. We will expand Sec. 3.6 to explain that SEAL’s robustness is bounded by the usual
>   “train‑your‑own‑model” asymmetry: if the attacker is willing to spend that level of effort, they can own a separate adapter,
>   but they still cannot *steal* the original owner’s passports.
>
> We will make this limitation explicit in the threat‑model / limitations sections, especially with respect to LoRA adapters trained on public datasets.
>
> ### (4) Placement of Spectral Diagnostics and relevance of white‑box watermarking
> We will:
> - move most of the detailed Spectral Diagnostics to the appendix or later in the experimental section, and
> - add 1–2 sentences up front explaining how the heavy‑tailed singular spectrum and rank behavior motivate
>   our obfuscation/ambiguity‑attack experiments.
>
> On white‑box vs. black‑box relevance: we will expand the discussion to emphasize that:
>
> - white‑box adapter watermarking is most relevant in ecosystems where adapters circulate as files, and
> - insights from SEAL may inform future black‑box schemes.
>
> Thank you again for the detailed comments—they directly helped us tighten both the policy and the exposition.

---

> ### Author Response · Authors · 2025-11-23
> **Revisions Uploaded**
>
> Thank you again for your constructive feedback. We have uploaded a revised manuscript that directly incorporates your suggestions.
>
> Summary of Changes for You:
> 1. **Strengthened Policy (R0)**: Explicitly added the "Structural Validity" check in Sec. 4.2 (Decision Rule Box) to reject trivial claims .
> 2. **Mistral-7B & Hoeffding**: Added a remark in Sec. 4.3 explaining why non-i.i.d. benchmarks may require empirical thresholds .
> 3. **Spectral Diagnostics**: Moved detailed analysis to Appendix to improve flow, keeping only the key motivation in Sec. 5.1 .
>
> To facilitate your check, we have highlighted these specific revisions in blue and marked them with the tag **d1zQ** in the right margin of the updated PDF. We hope these changes fully address your concerns.

---

> > ### Comment · Reviewer_d1zQ · 2025-11-27
> >
> > I thank the authors for the detailed and technically careful rebuttal and proposed revisions. The main concerns in my original review are largely addressed:
> >
> > - The introduction of an explicit policy-level precondition (R0) that enforces non-trivial passport separation and shape/rank/conditioning checks resolves the issue of trivial or identity-like passports being accepted under R1/R2.
> > - The discussion around Hoeffding’s inequality and the Mistral-7B case is now clearer: the non-i.i.d. nature of the commonsense suite explains why the bound fails there, and the rule that a formal FPR guarantee may only be claimed when the owner’s observed gap is below the Hoeffding threshold makes the distinction between “formal” and “empirical-only” acceptance explicit.
> > - The threat model and limitations in the open-data setting are now more clearly articulated: if the fine-tuning data are public and an attacker is willing to spend comparable compute, they can retrain their own SEAL adapter with their own passports. This is an inherent limitation and is now stated as such rather than being implicit; I would also emphasize that, since LoRA training is relatively cheap compared to base-model pretraining, this compute-asymmetry argument mainly protects against low-effort attackers rather than well-resourced ones.
> >
> > Some weaknesses remain, but I view them more as scope and evaluation limitations than fundamental flaws. In particular, I still believe it should be possible to adapt existing full-model white-box watermarking techniques for comparison, not only passport-like methods, and I do not fully agree with the argument given to reviewer WQP7 that such comparisons are out of scope. Nonetheless, I appreciate the overall presentation and empirical depth of the paper, I find the core mechanism (entangled passports with R0+R1+R2) conceptually sound as revised, and I keep my positive assessment of the work.

---

> > > ### Author Response · Authors · 2025-11-27
> > >
> > > We sincerely thank the reviewer for the continued support and for confirming that our revisions (R0 policy, theoretical clarifications, and threat model) have resolved the primary concerns.
> > >
> > > Regarding the remaining point on **comparisons with full-model watermarking**:
> > >
> > > We fully agree with your insight. As you pointed out, adapting existing white-box methods [1, 2] to LoRA is mechanically possible in principle—for instance, by applying weight regularization directly to the adapter matrices $B$ and $A$, or by inserting auxiliary layers.
> > >
> > > However, our decision to treat these as "out of scope" was driven by a specific design goal: **avoiding the training dynamics distortion** caused by auxiliary losses. Conventional watermarking losses often force a trade-off between markability and task performance (degradation). In contrast, SEAL's injection of fixed random/structured matrices allows for watermarking without auxiliary optimization objectives, and can even serve as a beneficial regularizer (similar to the benefits of fixed high-rank components discussed in [3]).
> > >
> > > Thus, while a direct comparison with [1, 2] is feasible **if one accepts the likely trade-off of reduced task fidelity**, we prioritized a **structurally native** approach that preserves the original LoRA's optimization trajectory and performance benefits. We will clarify this design rationale and the feasibility of adaptation in the final version.
> > >
> > > _References_
> > >
> > > [1] Uchida et al., "Embedding Watermarks into Deep Neural Networks", ICMR 2017.
> > >
> > > [2] Fan et al., "Rethinking Deep Neural Network Ownership Verification", NeurIPS 2019.
> > >
> > > [3] Albert et al., "RandLoRA: Full Rank Parameter-Efficient Fine-Tuning of Large Models", ICLR 2025.

---

### Author Response · Authors · 2025-11-14
**Global Clarification: A Strengthened Verification Policy (R0-R2)**

First of all, we sincerely thank all reviewers for the very detailed and technically sharp feedback.
Before addressing individual reviews, we would like to
(1) clarify our verification policy and how it rules out “trivial” or degenerate claims, and
(2) explain how conditioning and rank enter our security argument.

For points that require extra experiments or substantial edits to the main text, we will first respond here and then integrate the changes in a revised manuscript after we can incorporate all reviewers’ feedback coherently.

---

## From “R1+R2 only” to a strengthened policy: introducing R0

In the current draft, the public rule is phrased as:

- **R1 (reconstruction check).**
  Given a claim $(B,A,C,C_p)$, recompose the adapter $\Delta W(C)=BCA$ and compare it to the released adapter $(B',A')$.
  The *intended* rule is: $ \min_{C_i\in\{C, C_p\}} \| \Delta W(C_i) - B'A' \|_F\le\rho_T.$ We will correct the text in Sec. 4.2.
- **R2 (task fidelity / entanglement).**
  On a fixed evaluation suite $T$, measure $\Delta_T = \bigl| M_T\bigl(\mathbb{N}(\cdot, C)\bigr) - M_T\bigl(\mathbb{N}(\cdot, C_p)\bigr) \bigr|$ and accept if $\Delta_T \le \tau_T$.

Several reviewers rightly pointed out that, if taken in isolation, this leaves room for corner‑case claims such as
$(B,A)=(B',A')$ with $C=C_p=I$, or degenerate factorizations obtained by destroying conditioning via SVD truncation.

To close these *policy* gaps **without changing the SEAL training procedure itself**, we propose to make the following front‑end safeguard explicit in the public verification rule.

### R0 (policy‑level sanity check, applied before R1/R2)

A verifier rejects any claim $(B,A,C,C_p)$ that is *trivial or numerically degenerate*, via:

- **R0.a – non‑trivial passports.**
  Enforce a *minimum* matrix‑level separation $\|C - C_p\|_F > \delta_C$,   where $\delta_C$ is calibrated from the owner’s passport generation rule (e.g., structured bitmap vs. i.i.d. $\mathcal{N}(0,1)$; in that case the expected $\|C-C_p\|_F$ for $r\times r$ matrices (typically grows linearly with $r$ in expectation) then published as part of the policy. Identity‑like or nearly identical passports are rejected before any statistical test.

- **R0.b – induced weight separation.**
  Enforce that the two passports induce a non‑negligible difference in the adapter weights: $\|\Delta W(C) - \Delta W(C_p)\|_F \;>\; \kappa_0 \,\delta_C$, where $\kappa_0$ is a per‑adapter lower bound on the smallest singular value of the linear map $C \mapsto \Delta W(C)$ (see the lemma below). Intuitively, if two passports are too close *and* their induced weights are too close, they are not acceptable as distinct evidence of ownership.

- **R0.c – shape / rank / conditioning consistency.**
  Require that the submitted factors $(B,A)$:

  (i) have the **same shapes and ranks** as the released LoRA adapter $(B',A')$ (no extra truncation or expansion), and
  (ii) satisfy a minimal conditioning constraint $\\min_\\ell \\left\\{ \\sigma_{\\min}(B_\\ell),\\, \\sigma_{\\min}(A_\\ell) \\right\\} \\ge \\kappa_0,$
  where $\kappa_0$ is calibrated once on the released adapter (e.g., as the minimum over layers) and then fixed in the public policy.

Under this explicit R0, a would‑be attacker can no longer pass the public rule by
- submitting $C=C_p$ (or $C\approx C_p$)  $\Rightarrow$ caught by **R0.a / R0.b**;
- submitting a completely different low‑rank factorization with aggressive SVD truncation $\Rightarrow$ caught by **R0.c** (shape / rank mismatch and/or under‑conditioning).

We stress that this is a *policy refinement*, not a change to the SEAL scheme itself: the training objective, entanglement mechanism, and all experiments remain the same; we are simply stating more clearly what a *reasonable verifier* must reject.

---

### Conditioning, rank, and a lower bound on induced differences (informal lemma)
For each LoRA layer, the SEAL update is linear in the passport:
$\Delta W(C) = B C A.$ Vectorizing gives $\operatorname{vec}(\Delta W(C)) = (A^\top \otimes B)\,\operatorname{vec}(C),$
whose singular values are all pairwise products $\sigma_i(A)\sigma_j(B)$. Hence, if we define $\kappa_0 = \min_\ell \bigl(\sigma_{\min}(B_\ell)\,\sigma_{\min}(A_\ell)\bigr) > 0,$ then for any two passports $C, C_p$ we have the Frobenius‑norm lower bound $ \|\Delta W(C) - \Delta W(C_p)\|_F \ge \kappa_0 \,\|C - C_p\|_F, $ which is exactly what R0.b enforces at the policy level.

Empirically, for our Gemma-2B SEAL adapter on the commonsense benchmark, we measured across 90 LoRA layers
$\kappa_{\min} \approx 4.2\times 10^{-4}, \quad \kappa_{\text{mean}} \approx 4.2\times 10^{-3},$
providing a concrete, strictly positive lower bound for the deployed model. We will report these values (and the corresponding $\delta_C$) in the revised manuscript and make clear that they are task/model-specific hyper-parameters of the public verification policy rather than new training knobs.

---

### Author Response · Authors · 2025-11-28
**Summary of Key Rebuttal Updates**

Dear Area Chair and Reviewers,

As the discussion phase is approaching its end, we would like to provide a concise summary of our major updates to facilitate the final review process. We understand everyone has a busy schedule, but we would greatly appreciate your feedback on the revisions.

**1. Addressed Ambiguity & Security Concerns (For Reviewers P2Pv, WQP7)**
We have implemented a **Strengthened Verification Policy (R0)** in our Global Clarification and revised manuscript.
* **Mechanism:** This policy acts as a pre-condition sanity check that enforces non-trivial passport separation and conditioning.
* **Outcome:** It ensures that trivial (e.g., identity matrix) or degenerate passports are rejected *before* any statistical testing, effectively resolving the "Ambiguity Attack" concerns.

**2. Additional Robustness Experiments (For Reviewer fNRL)**
Following the suggestion, we conducted **Rank Ablation Studies ($r \in \{4, 8, 16\}$)**.
* **Outcome:** The results (Tables A & B) demonstrate that SEAL maintains consistent watermark detectability and task fidelity across different rank settings, confirming the method's scalability without unexpected trade-offs.

**3. Comparison and Scope (For Reviewer d1zQ)**
We have incorporated a discussion on the feasibility of adapting full-model watermarking methods into the final version. We explicitly acknowledge that while technically feasible, such adaptations may incur trade-offs regarding training dynamics that SEAL avoids by design.

We sincerely thank **Reviewer d1zQ** for the active engagement and constructive feedback, which has significantly strengthened the paper.

We remain available for any further questions and hope these updates fully address the remaining concerns of **Reviewers P2Pv, WQP7, and fNRL** before the deadline.

Best regards,

Authors

---

### Meta-Review · Area_Chair_6C3z · 2026-01-08

**Summary:**

The authors propose a white-box watermarking approach for LoRA adapters, addressing a relevant topic in the field. Reviewers appreciated the clarity, organization, and presentation of the ideas, noting the paper’s potential to make a meaningful contribution to watermarking. The proposed method is simple yet well-formulated, maintains the same inference complexity, and is supported by thorough experimental validation.

Reviewers raised concerns regarding contradictions between theoretical guarantees and empirical findings in the Mistral 7B case, the practical importance of the method in open-data settings, and its robustness under different attack scenarios. While most of these concerns were satisfactorily addressed in the rebuttal, a major issue remains: the approach’s vulnerability to ambiguity attacks. Specifically, R1 and R2 checks can be bypassed due to the non-uniqueness of matrix factorization. To address this, the authors proposed an additional step, R0, claiming it fully resolves the problem. However, the proposed correction requires significant and careful revision. Its correctness must be rigorously verified both theoretically and experimentally. For instance, it is unclear whether the lower bound on singular values can always be guaranteed to remain positive. Considering these unresolved issues, I recommend rejecting the paper at this stage and encourage the authors to refine their approach and resubmit to a future venue.

**Reviewer Concerns:**

Reviewers d1zQ and P2Pv raised concerns about the practical applicability of the method when models are trained on publicly available datasets. The authors addressed this issue and provided sufficient justification for their setup. They also responded to comments from reviewers fNRL and WQP7 regarding experimental validation, specifically the behavior of the approach under varying rank and complexity, comparisons with other watermarking methods, and detection of p-values.

However, a major concern raised by reviewers d1zQ and WQP7 involved the method’s vulnerability to ambiguity attacks. This issue revealed a technical flaw in the approach, which the authors attempted to address by introducing an additional step (R0).  Nevertheless, resolving this limitation requires more detailed investigation and stronger theoretical and experimental justification, which are not provided in the current version of the paper.

**Reviewer Scores:**

Reviewer d1zQ initially gave a score of 6, which they would most likely have maintained or increased after the rebuttal. Reviewers P2Pv and WQP7 gave scores of 2, and it is unlikely these would have been raised to an acceptance level (6 or higher) following the rebuttal. Finally, reviewer fNRL gave a score of 4; given that the authors addressed their comments to a large extent, it is likely this score would have been slightly increased to 6.

---

### Decision · Program_Chairs · 2026-01-26

Reject